# Forecasting in Offline Reinforcement Learning for Non-stationary Environments

**Suzan Ece Ada**[1,2]   **Georg Martius**[2]   **Emre Ugur**[1]   **Erhan Oztop**[3,4]

[1]Bogazici University, Türkiye   [2]University of Tübingen, Germany

[3]Ozyegin University, Türkiye   [4]Osaka University, Japan

ece.ada@bogazici.edu.tr

## Abstract

Offline Reinforcement Learning (RL) provides a promising avenue for training policies from pre-collected datasets when gathering additional interaction data is infeasible. However, existing offline RL methods often assume stationarity or only consider synthetic perturbations at test time, assumptions that often fail in real-world scenarios characterized by abrupt, time-varying offsets. These offsets can lead to partial observability, causing agents to misperceive their true state and degrade performance. To overcome this challenge, we introduce **F**orecasting in Non-stationary **O**ffline **RL** (FORL), a framework that unifies (i) conditional diffusion-based candidate state generation, trained without presupposing any specific pattern of future non-stationarity, and (ii) zero-shot time-series foundation models. FORL targets environments prone to unexpected, potentially non-Markovian offsets, requiring robust agent performance from the onset of each episode. Empirical evaluations on offline RL benchmarks, augmented with real-world time-series data to simulate realistic non-stationarity, demonstrate that FORL consistently improves performance compared to competitive baselines. By integrating zero-shot forecasting with the agent's experience, we aim to bridge the gap between offline RL and the complexities of real-world, non-stationary environments.

## 1   Introduction

Offline Reinforcement Learning (RL) leverages static datasets to avoid costly or risky online interactions [1, 2]. Yet, agents trained on fully observable states often fail when deployed with noisy or corrupted observations. While robust offline RL methods address test-time perturbations, such as sensor noise or adversarial attacks [3], a critical gap persists in addressing non-stationarity within the observation function—a challenge that fundamentally alters the agent's perception of the environment over time.

Prior *online algorithms* that consider the scope of non-stationarity as the observation function focus on learning agent morphologies [4] and generalization in Block MDPs [5]. While this scope of non-stationarity holds significant potential for real-world applications [6], it remains underexplored. We focus on the episodic evolution of the observation function at test-time in offline RL. In our setup, each dimen-

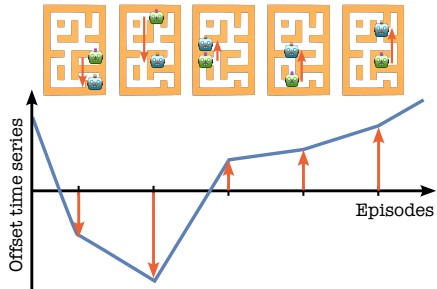

Figure 1: **Setting.** The agent does not know its location in the environment because its perception is offset every episode $j$ by an unknown offset $b^j$ (only vertical offsets are illustrated). FORL leverages historical offset data and offline RL data (from a stationary phase) to forecast and correct for new offsets at test time. Ground-truth offsets (↓,↑) are hidden throughout the evaluation episodes.

39th Conference on Neural Information Processing Systems (NeurIPS 2025).

sion of an agent's state is influenced by an unknown, time-dependent constant additive offset that remains fixed within a single operational interval (an "episode"). This leads to a stream of evolving observation functions [7], extending across multiple future episodes, **where the offsets remain hidden throughout the prediction window**. For instance, industrial robots might apply a daily calibration offset to each joint, while sensors can exhibit a deviation until the next scheduled recalibration. Similarly, in healthcare or finance, data may be partially aggregated or withheld to comply with regulations, effectively obscuring finer-grained variations and leaving a single offset as the dominant factor per episode. By only storing these representative offsets, we circumvent the challenges of continuous interaction buffers in bandwidth-constrained or privacy-sensitive environments. Because the offset can differ across state dimensions (e.g., different sensor or actuation channels), each state dimension can be affected by a different unknown bias that stays constant for that episode but evolves differently across episodes. Approaches that assume predefined perturbations can struggle with these abrupt, episodic shifts because such offsets violate the typical assumption of smoothly varying observation functions. Frequent retraining, hyperparameter optimization, or extensive online adaptation to new observation function evolution patterns are costly, risky (due to trial-and-error in safety-critical settings), and may be infeasible if these patterns no longer reflect assumptions made during training. By separating offset data (episodic calibration values) from the massive replay buffers, a zero-shot forecasting-based approach can anticipate each new offset from the beginning of the episode without requiring policy updates or making assumptions on task evolution at test time [8]. Modeling these multidimensional additive offsets as stable, per-episode constants presents a robust and efficient way to handle time-varying conditions in non-stationary environments where the evolution of tasks follows a non-Markovian, time-series pattern [9], mitigating the risks of online exploration.

We consider an offline RL setting during training where we have only access to a standard offline RL dataset collected from a stationary environment [3] with fully observable states. Initial data may be collected under near-ideal conditions and then gradually affected by wear, tear, or other natural shifts—even as the underlying physical laws (dynamics) remain unchanged. At test time, however, we evaluate in a non-stationary environment where both the observation function and the observation space change due to time-dependent external factors. This setup can be interpreted as environments shifting along observation space dimensions while the initial state of the agent is sampled from a uniform distribution over the state space. A simplified version of this setup for an offset affecting only one dimension of the state is illustrated in Figure 1. Here, the agent "knows" it is in a maze but does not know where it is in the maze. Furthermore, **it will remain uncertain of its location across all episodes at test-time**, as in every episode, a new offset leads to a systematic

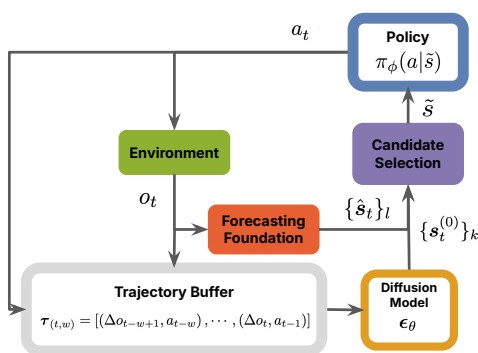

Figure 2: Overview of the proposed FORL framework at test-time. The observations are processed by both the trajectory buffer and the time-series **forecasting foundation** module [10]. Observation changes and actions sampled from the **policy** $(\Delta o, a)$, are stored in the trajectory buffer. The **diffusion model** generates candidate states $\{s_t^{(0)}\}_k$ conditioned on $\boldsymbol{\tau}_{(t,w)}$. The **candidate selection** module then generates the estimated $\tilde{s}_t$.

misalignment between perceived and actual positions. Importantly, these offsets may not conform to Gaussian or Markovian assumptions; instead, they may stem directly from complex, real-world time-series data [9] and remain constant throughout each episode. As a result, standard noise-driven or parametric state-estimation techniques, which typically rely on smoothly varying or randomly perturbed functions, cannot reliably identify these persistent, episode-wide offsets that are not available after the episode terminates. While zero-shot forecasting can adjust observation offsets, its performance depends on the forecaster's accuracy. Similarly, integrating zero-shot forecasting into a model-based offline RL approach [3] can underperform when real-world offsets deviate from predefined assumptions about future observation functions. Our approach uses the insight that the belief of the true states can be refined from a sequence of actions and effects. For instance, in maze navigation, if an agent misjudges its location and hits a wall, analyzing its actions and delta observations leading to the collision can provide evidence for likely locations within the maze.

We propose the **F**orecasting in Non-stationary **O**ffline **RL** (FORL) framework (Figure 2) for test-time adaptation in non-stationary environments where the observation function is perturbed by an arbitrary time-series. Our framework has two main ingredients: forecasting offsets with a zero-shot time-series forecasting model [10] from past episode offsets (ground truth offsets after the episode terminates are not accessible at test-time) and a within-episode update of the state estimation using a conditional diffusion model [11] trained on offline stationary data.

**Contributions.** We unify the strengths of probabilistic forecasting and decision-making under uncertainty to enable continuous adaptation when the environment diverges from predictions. Consequently, our framework: *(1)* accommodates future offsets *without assuming specific non-stationarity patterns during training*, eliminating the need for retraining and hyperparameter tuning when the agent encounters new, unseen non-stationary patterns at test time, and *(2) targets non-trivial non-stationarities at test time without requiring environment interaction or knowledge of POMDPs during training*. *(3)* FORL introduces a novel, modular framework combining a conditional diffusion model (FORL-DM) for multimodal belief generation with a lightweight Dimension-wise Closest Match (DCM) fusion strategy, validated by extensive experiments on no-access to past offsets, policy-agnostic plug-and-play, offset magnitude-scaling, and inter/intra-episode drifts. *(4)* We propose a novel benchmark that integrates offsets from real-world time-series datasets with standard offline RL benchmarks and demonstrate that FORL consistently outperforms baseline methods.

**Background: Diffusion Models** Denoising diffusion models [12, 13] aim to model the data distribution with $p_\theta(\boldsymbol{x}_0) := \int p_\theta(\boldsymbol{x}_{0:N}) \, d\boldsymbol{x}_{1:N}$ from samples $x_0$ in the dataset. The joint distribution follows the Markov Chain $p_\theta(\boldsymbol{x}_{0:N}) := \mathcal{N}(\boldsymbol{x}_N; \mathbf{0}, \mathbf{I}) \prod_{n=1}^{N} p_\theta(\boldsymbol{x}_{n-1} \mid \boldsymbol{x}_n)$ where $\boldsymbol{x}_n$ is the noisy sample for diffusion timestep $n$ and $p_\theta(\boldsymbol{x}_{n-1} \mid \boldsymbol{x}_n) := \mathcal{N}(\boldsymbol{x}_{n-1}; \boldsymbol{\mu}_\theta(\boldsymbol{x}_n, n), \boldsymbol{\Sigma}_\theta(\boldsymbol{x}_n, n))$. During training, we use the samples from the distribution $q(\boldsymbol{x}_n \mid \boldsymbol{x}_0) = \mathcal{N}(\boldsymbol{x}_n; \sqrt{\bar{\alpha}_n} \boldsymbol{x}_0, (1 - \bar{\alpha}_n) \mathbf{I})$ where $\bar{\alpha}_n = \prod_{i=1}^{n} \alpha_i$ [11]. General information on diffusion models is given in Section B.3.

# 2 Method

In this section, we formulate our problem statement and describe our FORL diffusion model trained on the offline RL dataset to predict plausible states. Then, we introduce our online state estimation method, Dimension-wise Closest Match that uses plausible states predicted by the multimodal FORL diffusion model (DM) and the states predicted from past episodes prior to evaluation by a probabilistic unimodal zero-shot time-series foundation model.

## 2.1 Problem Statement

**Training (Offline Stationary MDP)** We begin with an episodic, stationary Markov Decision Process (MDP) $\mathcal{M}_{\text{train}} = (\mathcal{S}, \mathcal{A}, \mathcal{T}, \mathcal{R}, \rho_0)$, where the initial state distribution $\rho_0$ is a uniform distribution over the state space $\mathcal{S}$. We only have access to an offline RL dataset $\mathcal{D} = \{(\boldsymbol{s}_t^k, \boldsymbol{a}_t^k, \boldsymbol{s}_{t+1}^k, r_t^k)\}$ with $k$ transitions collected from this MDP. Crucially, our FORL diffusion model and a diffusion policy [14] are trained offline using this dataset, such as the standard D4RL benchmark [15], without making any assumptions about how the environment might become non-stationary at test time.

**Test Environment (Sequence of POMDPs)** At test time, the agent faces an infinite sequence of POMDPs $\{\hat{\mathcal{M}}_j\}_{j=1}^{\infty}$. Each POMDP is described by a 7-tuple [16] $\hat{\mathcal{M}}_j = (\mathcal{S}, \mathcal{A}, \mathcal{O}_j, \mathcal{T}, \mathcal{R}, \rho_0, \mathbf{x}_j)$, where the state space $\mathcal{S}$, action space $\mathcal{A}$, transition function $\mathcal{T}$, and the reward function $\mathcal{R}$ remain identical to the training MDP. $\mathbf{x}$ is the observation function, where we restrict ourselves to deterministic versions ($\mathbf{x} : \mathcal{S} \to \mathcal{O}$) [6, 17]. Non-stationary environments can be formulated in different ways. In Khetarpal et al. [6], a *general non-stationary RL* formulation is put forward, which allows each component of the underlying MDP or POMDP to evolve over time, i.e. $(\mathcal{S}(t), \mathcal{A}(t), \mathcal{T}(t), \mathcal{R}(t), \mathbf{x}(t), \mathcal{O}(t))$. A set $\kappa$ specifies which of these components vary, and a *driver* determines how they evolve. In particular, *passive* drivers of non-stationarity imply that exogenous factors alone govern the evolution of the environment, independent of the agent's actions. In this work, we consider the scope of non-stationarity [6] (Section B.2) to be the observation function and the observation space, i.e. $\kappa = \{\mathbf{x}, \mathcal{O}\}$.

We consider the case where non-stationarity unfolds over episodes and where the observation function $\mathbf{x}_j$ is different in each episode $j$. The change in the observation function is assumed to have an

additive structure and is independent of the agent's actions (passive non-stationarity [6]). Concretely, the function $\mathbf{x}_j$ offsets states $s_t$ by a fixed offset $b^j \in \mathbb{R}^n$:

$$\mathcal{O}_j = \{s + b^j : s \in \mathcal{S}\}, \quad \mathbf{x}_j(s) = s + b^j.$$

Importantly, the sequence $(b^j)$ can evolve under arbitrary real-world time-series data, and the agent **does not have access to the ground-truth information throughout the evaluation**—similar to scenarios where observations are only available periodically and shifts occur between these intervals. Thus, the episodes have a temporal order, relating to Non-Stationary Decision Processes (NSDP), defining a sequence of POMDPs [7] (Section B.2).

**Partial Observability and Historical Context**   Since $b^j$ is **never directly observed** for $P$ episodes into the future, each $\hat{\mathcal{M}}_j$ is a POMDP. The agent receives only the offset-shifted observations $(o_t)$, where $o_t = s_t + b^j$ without any noise. Moreover, the agent may have access to a limited historical context of previous offsets $(b^{j-C}, \ldots, b^{j-1})$ at discrete intervals $P$, but **no direct information** about future offsets $(b^j, b^{j+1}, \ldots b^{j+P-1})$. Hence, the agent must forecast and/or adapt to unknown future offsets without prior non-stationary training.

**Partial Identifiability**   Despite observing $o_t = s_t + b^j$, the agent cannot generally disentangle $s_t$ from $b^j$. For any single observation, there are infinite possible pairs of state and offset that yield $o_t = s' + b'$. Additionally, the initial state distribution $\rho$ is uniform and does not provide information about $b$. Thus, we can only form a belief over $s_t$ and refine that belief based on two sources of information: a) the sequence of actions and effects observed within an episode and b) the sequence of past identified offsets. To exploit source a, we utilize a predictive model of commonly expected outcomes based on a diffusion model, which will be explained next. To make use of source b, we use a zero-shot forecasting model (see Section 2.3 for details). Afterwards, both pieces of information are fused (Section 2.4).

## 2.2   FORL Diffusion Model

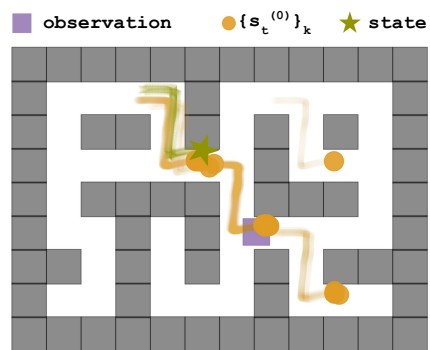

Figure 3: Candidate states generated by FORL Diffusion Model.

In our setting, we assume that the offsets added to the states are unobservable at test time, while the transition dynamics of the evaluation environment remain unchanged. To eventually reduce the uncertainty about the underlying state, we perform filtering or belief updates using the sequence of past interactions. To understand why the history of interactions is indicative of a particular ground truth state, consider the following example in a maze environment illustrated in Fig. 3. When the agent moves north for three steps and then bumps into a wall, the possible ground truth states can only be those three steps south of any wall. The agent cannot observe the hidden green trajectory of ground-truth states; it only has access to the sequence of observation changes $(\Delta o)$ and action vectors, which narrows down the possible positions to four candidate regions—exactly those identified by our model. Clearly, the distribution of possible states is highly multimodal, such that we propose using a diffusion model as a flexible predictive model of plausible states given the observed actions and outcomes. Diffusion models excel at capturing multimodal distributions [14], making them well-suited for our task. We train the diffusion model on offline data without offsets $(b_j = 0)$ which we detail below.

To distinguish between the trajectory timesteps in reinforcement learning (RL) and the timesteps in the diffusion process, we use the subscript $t \in \{0, \ldots, T\}$ to refer to RL timesteps and $n \in \{0, \ldots, N\}$ for diffusion timesteps. We first begin by defining a subsequence of a trajectory

$$\boldsymbol{\tau}_{(t,w)} = [(\Delta o_{t-w+1}, a_{t-w}), \cdots, (\Delta o_t, a_{t-1})]. \tag{1}$$

where delta observations $\Delta o_t = o_t - o_{t-1} = s_t - s_{t-1}$ denote the state changes (effects), $w$ is the window size. Using a conditional diffusion model, we aim to model the distribution $p(\boldsymbol{s}_t \mid \boldsymbol{\tau}_{(t,w)})$. For that, we define the reverse process (denoising) as

$$p(s_t^{(N)}) \prod_{n=1}^{N} p_\theta\left(\boldsymbol{s}_t^{(n-1)} \mid \boldsymbol{s}_t^{(n)}, \boldsymbol{\tau}_{(t,w)}\right), \quad p(s_t^{(N)}) = \mathcal{N}(\boldsymbol{s}_t^{(N)}; \mathbf{0}, \mathbf{I}) \tag{2}$$

and $p_\theta$ is modeled as the distribution $\mathcal{N}(s_t^{(n-1)}; \mu_\theta(s_t^{(n)}, \tau_{(t,w)}, n), \Sigma_\theta(s_t^{(n)}, \tau_{(t,w)}, n))$ with a learnable mean and variance. We could directly supervise the training of $\mu_\theta$ using the forward (diffusion) process. Following Ho et al. [11], Song et al. [18], we compute a noisy sample $s_t^{(n)}$ based on the true sample $s_t = s_t^{(0)}$:

$$s_t^{(n)} = \sqrt{\bar{\alpha}(n)} s_t + \sqrt{1 - \bar{\alpha}(n)} \epsilon \tag{3}$$

where $\epsilon \sim \mathcal{N}(\mathbf{0}, \mathbf{I})$ is the noise, $\bar{\alpha}(n) = \prod_{i=1}^{n} \alpha(i)$ and the weighting factors $\alpha(n) = e^{-\left(\beta_{\min}\left(\frac{1}{N}\right) + (\beta_{\max} - \beta_{\min})\frac{2n-1}{2N^2}\right)}$ where $\beta_{\max} = 10$ and $\beta_{\min} = 0.1$ are parameters introduced for empirical reasons [19].

We can equally learn to predict the true samples by learning a noise model [20]. Hence, we train a noise model $\epsilon_\theta(s_t^{(n)}, \tau_{(t,w)}, n)$ that learns to predict the noise vector $\epsilon$. By using the conditional version of the simplified surrogate objective from [11], we minimize

$$\mathcal{L}_p(\theta) = \mathbb{E}_{n,\tau,s_t,\epsilon} \left[ \left\| \epsilon - \epsilon_\theta\left(s^{(n)}, \tau_{(t,w)}, n\right) \right\|^2 \right] \tag{4}$$

where $s_t$ is the state sampled from the dataset $D$ for $t \sim U_T(\{w, \ldots, T-1\})$, $s^{(n)}$ is computed according to Eq. (3), $\epsilon$ is the noise, and $n \sim U_D(\{1, \ldots, N\})$ is the uniform distribution used for sampling the diffusion timestep.

We use the true data sample $s_t$ from the offline RL dataset to obtain the noisy sample in Eq. (3). Leveraging our model's capacity to learn multimodal distributions, we generate a set of $k$ samples $\{s_t^{(0)}\}$ as our **predicted state candidates** in parallel from the reverse diffusion chain. We use the noise prediction model [11] with the reverse diffusion chain $s_t^{(n-1)} \mid s_t^{(n)}$ formulated as

$$\frac{s_t^{(n)}}{\sqrt{\alpha_{(n)}}} - \frac{1 - \alpha_{(n)}}{\sqrt{\alpha_{(n)}(1 - \bar{\alpha}_{(n)})}} \epsilon_\theta(s_t^{(n)}, \tau_{(t,w)}, n) + \sqrt{1 - \alpha_{(n)}} \epsilon \tag{5}$$

where $\epsilon \sim \mathcal{N}(0, \mathbf{I})$ for $n = N, \ldots, 1$, and $\epsilon = 0$ for $n = 1$ [11]. Below, we detail how the state candidates are used during an episode.

## 2.3 Forecasting using Zero-Shot Foundation Model

Because we assume that the offsets $b^j$ originate from a time series, we propose using a probabilistic zero-shot forecasting foundation model *(Zero-Shot FM)* such as Lag-Llama [10], to forecast future offsets from past ones. We assume that after $P$ episodes, the true offsets are revealed, and we predict the offsets for the following $P$ episodes. Using the probabilistic *Zero-Shot FM* we generate $(\hat{b}_l^j, \ldots, \hat{b}_l^{j+P-1})$, where $(l)$ denotes the number of samples generated for each episode (timestamp). Since Lag-Llama is a probabilistic model, it can generate multiple samples per timestamp, conditioned on $C$ number of past contexts $(b^{j-C}, \ldots, b^{j-1})$. In practice, we forecast every dimension of $b$ independently since the *Zero-Shot FM* (Lag-Llama [10]) is a univariate probabilistic model.

**Algorithm 1** Candidate Selection

1: Sample $\{\{\hat{b}\}_l^1, \ldots, \{\hat{b}\}_l^P\} \sim$ *Zero-Shot FM*
2: **for** each episode $p = 1, \cdots, P$ **do**
3:     $t = 0$
4:     Reset environment $o_0 \sim \mathcal{E}$
5:     $\tilde{s} \leftarrow o_0 - \overline{\{\hat{b}\}_l^p}$
6:     Initialize $\tau_{(t,w)}$
7:     **while** not *done* **do**
8:         Sample $a^{(0)} \sim \pi_\phi(a|\tilde{s})$
9:         Take action $a^{(0)}$ in $\mathcal{E}$, observe $o_{t+1}$
10:         $\{\hat{s}_{t+1}^b\}_l \leftarrow o_{t+1} - \{\hat{b}\}_l^p$
11:         $\tau_{(t+1,w)} = \text{PUSH}\left(\tau_{(t,w)}, \left(\Delta o_{t+1}, a^{(0)}\right)\right)$
12:         $\tau_{(t+1,w)} = \text{POP}\left(\tau_{(t+1,w)}, \left(\Delta o_{t-w+1}, a_{t-w}\right)\right)$
13:         **if** $t > w$ **then**
14:             Sample $\{s_{t+1}^{(0)}\}_k$ from FORL by Eq. 5
15:             $\tilde{s} \leftarrow \text{DCM}(\{s_{t+1}^{(0)}\}_k, \{\hat{s}_{t+1}^b\}_l)$
16:         **else**
17:             $\tilde{s} \leftarrow o_{t+1} - \overline{\{\hat{b}\}_l^p}$
18:         **end if**
19:         $t \leftarrow t + 1$
20:     **end while**
21: **end for**

## 2.4 FORL State Estimation

The next step in our method is to fuse the information from the forecaster and the diffusion model into a state estimate used for control at test time.

At the beginning of an episode, no information can be obtained from the diffusion model, so for the first $w$ steps we only rely on the forecaster's mean prediction, i.e. $\tilde{s}_t = o_t - \overline{\hat{b}^j}$ where the mean is taken over the $l$ samples.

As soon as $w$ steps are taken, our FORL *State Estimation* improves on the inferred state as detailed below. Figure 2 offers an overview of the entire system and Algorithm 1 provides a detailed pseudocode.

To recap, the diffusion model generates samples $\{\boldsymbol{s}_t^{(0)}\}_k$ from the in-episode history $\tau$, Eq. (1). These samples represent a multimodal distribution of plausible state regions. The *Zero-Shot FM* generates $l$ samples of offsets $\{\hat{b}\}_l$ from which we compute forecasted states using $\{\hat{s}_t\}_l = o_t - \{\hat{b}\}_l$.

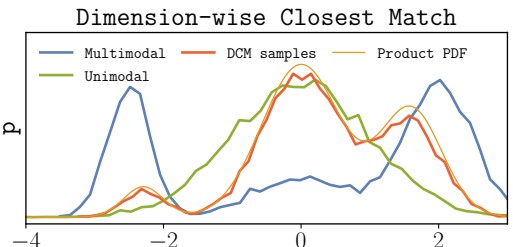

Figure 4: Distribution of samples produced by DCM (histograms for 10k samples for illustration).

**FORL: Dimension-wise Closest Match (DCM)**   We propose a lightweight approach to sample a good estimate based on the samples from the multimodal (*diffusion model* $\{\boldsymbol{s}_t^{(0)}\}_k$) and unimodal (*Zero-Shot FM* $\{\hat{s}_t^b\}_l$) distributions. Let $\mathcal{D}_{\text{diffusion}} = \{\mathbf{x}_1, \ldots, \mathbf{x}_k\}$, $\mathcal{D}_{\text{timeseries}} = \{\mathbf{y}_1, \ldots, \mathbf{y}_l\}$, where $\mathbf{x}_i, \mathbf{y}_j \in \mathbb{R}^n$. Then DCM constructs $\mathbf{z} \in \mathbb{R}^n$ by

$$z_d = y_{j^*(d),d} \quad \text{where} \quad j^*(d) = \arg\min_j \Big( \min_i \big| x_{i,d} - y_{j,d} \big| \Big),$$

where $d = 1 \ldots n$. In other words, for each dimension $d$, we choose the sample from $\mathcal{D}_{\text{timeseries}}$ that has the closest sample in $\mathcal{D}_{\text{diffusion}}$. The process is straightforward yet effective, and under ideal sampling conditions for a toy dataset in Fig. 4, DCM approximately samples from the product distribution. DCM uses a non-parametric search to find the forecast sample with the highest score, which corresponds to the minimum dimension-wise distance. DCM's prediction error is governed by the accuracy of the forecast samples in the unimodal $\mathcal{D}_{\text{timeseries}}$ that achieves this best score. As we will demonstrate in the experiments, this approach empirically yields lower maximum errors and is more stable compared to other methods.

**FORL Algorithm**   Algorithm 1 summarizes the entire inference process at test time. We begin the episode by relying on the forecasted states $\tilde{s}_0$. As more transitions $(\Delta o_t, a_{t-1})$ become available, the FORL diffusion model proposes candidate states $\{\boldsymbol{s}_t^{(0)}\}_k$ through *retrospection*—reasoning over the past in-episode experience to adapt state estimation on the fly when they begin to diverge from predictions. We then invoke DCM to blend the diffusion model's candidates with the foundation model's unimodal forecasts and obtain the final state estimate $\tilde{s}_t$. We use an off-the-shelf offline RL policy such as Diffusion-QL (DQL) [14] to select the agent's action $a_t$.

**Summary**   By combining a powerful *zero-shot* forecasting model with a *conditional diffusion* mechanism, FORL addresses partial observability in continuous state and action space when ground-truth offsets are unavailable. This procedure is performed in the *absence of ground-truth offsets for past, current, and future episodes over the interval $j : j+P$ at test time*. DCM provides a computationally inexpensive yet effective way of using the multimodal diffusion candidates and unimodal time-series forecasts. This robust adaptation approach yields a state estimate $\tilde{s}_t$, aligned with the agent's retrospective experience in the stationary offline RL dataset, incorporating a prospective external offset forecast.

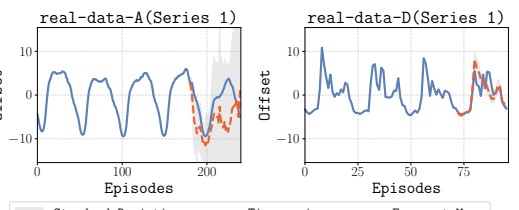

Figure 5: Zero-shot forecasting results of Lag-Llama [10] for the first univariate series (plotted) from the `real-data-A,D` datasets; experiments use the **first two series from each dataset**.

## 3 Experiments

We evaluate FORL across navigation and manipulation tasks in D4RL [15] and OGBench [21] offline RL environments, each augmented with five real-world non-stationarity domains sourced from [22]. Fig. 5 presents the ground truth, forecast mean, and standard deviation from Lag-Llama [10] for the *first series* of `real-data-A` and `real-data-D`. Our experiments address the following questions: **(1)** Does FORL maintain state-of-the-art performance when confronted with unseen non-stationary offsets? **(2)** How can we use FORL when we have no access to delayed past ground truth offsets? **(3)** How does DCM compare to other fusion approaches? **(4)** Can FORL handle intra-episode non-stationarity? **(5)** How gracefully does performance degrade as offset magnitude $\alpha$ is scaled from 0 (no offset) $\rightarrow$ 1 (our evaluation setup)? **(6)** Can FORL serve as a plug-and-play module for different offline RL algorithms without retraining? Extended results, forecasts for the remaining series, and implementation details are provided in the Appendix. Results average 5 seeds, unless noted.

**Baselines** We compare our approach with the following baselines: DQL [14], Flow Q-learning (FQL) [23] are diffusion-based and flow-based offline RL policies, respectively, that do not incorporate forecast information. DQL+LAG-$\bar{s}$, FQL+LAG-$\bar{s}$ extend DQL and FQL by using the sample mean of the forecasted states $\{\hat{s}_t\}_l$ at each timestep (using the constant per-episode predicted $b^j$). DQL+LAG-$\tilde{s}$ similarly extends DQL using the median. DMBP+LAG is a variant of the Diffusion Model–Based Predictor (DMBP)[3] (a robust offline RL algorithm designed to mitigate state-observation perturbations at test time, detailed in Appendix D) that integrates forecasted states from *Zero-Shot FM* [10] into its state prediction module. By using the model learned from the offline data, DMBP+LAG aims to refine the forecasted states to make robust state estimations. The underlying policies throughout our experiments are identical policy checkpoints for both our method and the baselines.

**Illustrative Example** Figs. 6 and 16 illustrate an agent navigating the `maze2d-large` environment where the true position is labeled as "state". The agent receives an observation indicating where it *believes* it is located due to unknown time-dependent factors. The candidate states predicted by the FORL diffusion model are shown as circles. Importantly, the agent's $(\Delta o, a)$-trajectory can reveal possible states for the agent. FORL's diffusion model (DM) compo-

Table 1: **Normalized scores (mean ± std.) for FORL framework and the baselines.** Bold are the best values, and those not significantly different ($p > 0.05$, Welch's t-test).

| maze2d-medium | DQL | DQL+LAG-$\bar{s}$ | DMBP+LAG | FORL (ours) |
|---|---|---|---|---|
| real-data-A | $30.2 \pm 6.5$ | $30.2 \pm 8.6$ | $25.1 \pm 9.8$ | $\mathbf{63.3} \pm 6.7$ |
| real-data-B | $14.1 \pm 12.1$ | $53.4 \pm 14.6$ | $41.2 \pm 21.1$ | $\mathbf{66.5} \pm 18.2$ |
| real-data-C | $-2.3 \pm 3.3$ | $56.7 \pm 18.5$ | $56.9 \pm 18.4$ | $\mathbf{86.3} \pm 15.7$ |
| real-data-D | $4.7 \pm 5.0$ | $36.9 \pm 16.3$ | $38.5 \pm 14.2$ | $\mathbf{103.4} \pm 11.9$ |
| real-data-E | $3.5 \pm 8.8$ | $8.7 \pm 6.0$ | $11.4 \pm 2.8$ | $\mathbf{51.2} \pm 13.7$ |
| **Average** | 10.0 | 37.2 | 34.6 | **74.1** |
| maze2d-large | | | | |
| real-data-A | $16.2 \pm 5.5$ | $2.4 \pm 1.1$ | $4.2 \pm 5.8$ | $\mathbf{42.9} \pm 4.1$ |
| real-data-B | $-0.5 \pm 2.9$ | $5.5 \pm 9.0$ | $15.0 \pm 14.6$ | $\mathbf{34.9} \pm 9.2$ |
| real-data-C | $0.9 \pm 1.7$ | $16.6 \pm 7.5$ | $26.8 \pm 8.4$ | $\mathbf{45.6} \pm 4.1$ |
| real-data-D | $3.0 \pm 6.6$ | $8.6 \pm 3.2$ | $13.4 \pm 4.1$ | $\mathbf{58.4} \pm 6.5$ |
| real-data-E | $-2.1 \pm 0.4$ | $\mathbf{2.6} \pm 3.4$ | $\mathbf{0.9} \pm 3.7$ | $\mathbf{12.0} \pm 9.9$ |
| **Average** | 3.5 | 7.1 | 12.1 | **38.8** |
| antmaze-umaze-diverse | | | | |
| real-data-A | $22.7 \pm 3.0$ | $41.0 \pm 5.2$ | $45.7 \pm 4.8$ | $\mathbf{65.3} \pm 8.7$ |
| real-data-B | $24.2 \pm 3.5$ | $48.3 \pm 7.0$ | $62.5 \pm 13.2$ | $\mathbf{74.2} \pm 10.8$ |
| real-data-C | $21.7 \pm 3.5$ | $50.4 \pm 8.3$ | $60.4 \pm 3.9$ | $\mathbf{78.8} \pm 8.5$ |
| real-data-D | $5.8 \pm 2.3$ | $26.7 \pm 6.3$ | $29.2 \pm 5.9$ | $\mathbf{75.8} \pm 8.0$ |
| real-data-E | $6.0 \pm 6.8$ | $58.0 \pm 16.6$ | $59.3 \pm 7.6$ | $\mathbf{81.3} \pm 6.9$ |
| **Average** | 16.1 | 44.9 | 51.4 | **75.1** |
| antmaze-medium-diverse | | | | |
| real-data-A | $31.0 \pm 6.5$ | $\mathbf{40.0} \pm 5.7$ | $\mathbf{39.7} \pm 4.0$ | $\mathbf{44.0} \pm 7.9$ |
| real-data-B | $23.3 \pm 4.8$ | $\mathbf{48.3} \pm 4.8$ | $43.3 \pm 16.0$ | $\mathbf{55.8} \pm 7.0$ |
| real-data-C | $10.0 \pm 2.3$ | $48.3 \pm 3.4$ | $49.6 \pm 3.7$ | $\mathbf{52.9} \pm 9.5$ |
| real-data-D | $11.7 \pm 5.4$ | $46.7 \pm 7.5$ | $41.7 \pm 6.6$ | $\mathbf{64.2} \pm 8.6$ |
| real-data-E | $\mathbf{18.7} \pm 4.5$ | $27.3 \pm 8.6$ | $\mathbf{26.0} \pm 5.5$ | $\mathbf{26.7} \pm 4.7$ |
| **Average** | 18.9 | 42.1 | 40.1 | **48.7** |
| antmaze-large-diverse | | | | |
| real-data-A | $11.0 \pm 1.9$ | $11.3 \pm 4.9$ | $9.0 \pm 4.5$ | $\mathbf{34.3} \pm 5.7$ |
| real-data-B | $5.8 \pm 4.8$ | $9.2 \pm 4.6$ | $8.3 \pm 2.9$ | $\mathbf{46.7} \pm 11.9$ |
| real-data-C | $5.4 \pm 2.4$ | $22.1 \pm 5.6$ | $17.9 \pm 3.8$ | $\mathbf{33.8} \pm 6.8$ |
| real-data-D | $2.5 \pm 2.3$ | $14.2 \pm 3.7$ | $14.2 \pm 6.3$ | $\mathbf{46.7} \pm 12.6$ |
| real-data-E | $\mathbf{5.3} \pm 3.8$ | $\mathbf{3.3} \pm 2.4$ | $\mathbf{3.3} \pm 0.0$ | $\mathbf{11.3} \pm 7.3$ |
| **Average** | 6.0 | 12.0 | 10.5 | **34.6** |
| kitchen-complete | | | | |
| real-data-A | $\mathbf{16.6} \pm 1.4$ | $7.2 \pm 1.9$ | $8.7 \pm 1.3$ | $12.0 \pm 3.9$ |
| real-data-B | $12.9 \pm 4.1$ | $32.7 \pm 6.5$ | $20.0 \pm 3.1$ | $\mathbf{33.1} \pm 5.6$ |
| real-data-C | $13.4 \pm 1.7$ | $23.9 \pm 6.6$ | $20.5 \pm 3.3$ | $23.9 \pm 6.0$ |
| real-data-D | $7.5 \pm 2.5$ | $24.0 \pm 9.2$ | $28.1 \pm 8.1$ | $27.1 \pm 10.1$ |
| real-data-E | $\mathbf{18.5} \pm 6.0$ | $2.8 \pm 2.1$ | $6.2 \pm 1.7$ | $10.3 \pm 3.0$ |
| **Average** | 13.8 | 18.1 | 16.7 | 21.3 |
| cube-single-play | FQL | FQL+LAG-$\bar{s}$ | | FORL-F (ours) |
| real-data-A | $0.0 \pm 0.0$ | $0.0 \pm 0.0$ | | $\mathbf{23.7} \pm 3.6$ |
| real-data-B | $0.0 \pm 0.0$ | $15.0 \pm 7.0$ | | $\mathbf{60.0} \pm 7.0$ |
| real-data-C | $0.4 \pm 0.9$ | $10.0 \pm 1.7$ | | $\mathbf{42.1} \pm 5.6$ |
| real-data-D | $0.0 \pm 0.0$ | $0.8 \pm 1.9$ | | $\mathbf{70.0} \pm 13.0$ |
| real-data-E | $0.0 \pm 0.0$ | $0.0 \pm 0.0$ | | $\mathbf{32.7} \pm 9.5$ |
| **Average** | 0.1 | 5.2 | | **45.7** |
| antmaze-large-navigate | | | | |
| real-data-A | $\mathbf{22.7} \pm 2.2$ | $1.3 \pm 0.7$ | | $\mathbf{24.3} \pm 4.3$ |
| real-data-B | $21.7 \pm 5.4$ | $29.2 \pm 8.8$ | | $\mathbf{40.0} \pm 7.6$ |
| real-data-C | $5.0 \pm 1.1$ | $34.6 \pm 6.7$ | | $\mathbf{55.8} \pm 3.7$ |
| real-data-D | $0.8 \pm 1.9$ | $37.5 \pm 5.1$ | | $\mathbf{75.8} \pm 5.4$ |
| real-data-E | $\mathbf{10.0} \pm 4.1$ | $3.3 \pm 0.0$ | | $15.3 \pm 8.7$ |
| **Average** | 12.0 | 21.2 | | **42.2** |

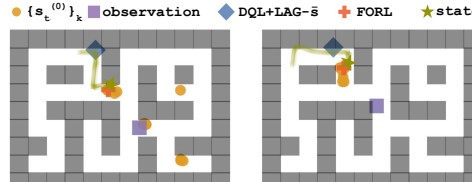

Figure 6: Visualization of states, predicted states as the agent navigates the environment.

nent predicts these candidate states by using observation changes ($\Delta o$) and corresponding actions ($a$). The possible candidate regions where the agent can be are limited, and our model successfully captures these locations. FORL's candidate selection module (DCM) uses the samples from the forecaster and the diffusion model to recover a close estimate for the state. In contrast, the baseline DQL+LAG-$\bar{s}$ relies on the forecaster [10] for state predictions, which are significantly farther from the actual state. Consistent with the results in Fig. 7, FORL reduces prediction errors at test-time, thereby improving performance.

### 3.1 Results

FORL outperforms both pure forecasting (DQL+LAG-$\bar{s}$) and the two-stage strategy that first predicts offsets with a time-series model and then applies a noise-robust offline RL algorithm (DMBP+LAG). Its advantage is consistent across previously unseen non-stationary perturbations from five domains, each introducing a distinct univariate series into a separate state dimension at test time. We present the average normalized scores over the prediction length $P$ across multiple

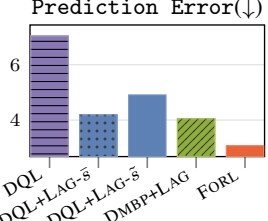

Figure 7: Prediction Error in recovering true agent state.

episodes run in the D4RL [15] and OGBench [21] for each time-series in Table 1. We conduct pairwise Welch's t-tests across all settings. Figure 7 plots the $\ell_2$ norm between the ground-truth states $s_t$ and those predicted by FORL and the baselines in the `antmaze` and `maze2d` environments. Consistent with the average scores, FORL achieves the lowest prediction error on average.

#### 3.1.1 No Access to Past Offsets

We evaluate different variants of using DM and *Zero-Shot FM* when we do not have any access to past offsets in Fig. 8. **FORL-DM (DM):** Diffusion Model utilizes the candidate states generated by the FORL's diffusion model component (Section 2.2), which can be a multimodal distribution (Fig. 3). Compared to DM, the full FORL framework yields a 97.8% relative performance improvement. Notably, DM performs on par with our extended baselines that incorporate historical offsets and forecasting—DMBP+LAG, DQL+LAG-$\bar{s}$,

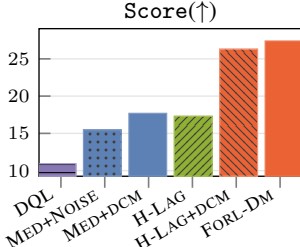

Figure 8: DM Ablations

and DQL+LAG-$\tilde{s}$. Moreover, without access to historical offset information before evaluation, DM achieves a 151.4% improvement over DQL, demonstrating its efficacy as a standalone module trained solely on a standard, stationary offline RL dataset without offset labels. **H-LAG:** We maintain a history of offsets generated by DM over the most recent $C$ episodes (excluding the evaluation interval $P$, since offsets are not revealed after episode termination at test-time). We then feed this history into the *Zero-Shot FM* to generate offset samples for the next $P$ evaluation episodes. These samples are applied directly at test time. **H-LAG+DCM:** We initially follow the same procedure in H-LAG to obtain predictions from *Zero-Shot FM*. Then, we apply **DCM** to these predicted offsets and the candidate states generated by FORL's diffusion model. We also compare against MED+DCM and MED+NOISE, simpler median-based heuristics detailed in Section G. Empirically, H-LAG+DCM outperforms H-LAG, demonstrating that DCM with FORL's diffusion model can improve robustness. Overall, scores and prediction errors indicate that just using the samples from DM has better scores on average, while H-LAG+DCM is more stable in Fig. 15.

#### 3.1.2 Dimension-wise Closest Match (DCM) Ablations

We compare FORL (DCM) against four alternative fusion strategies. FORL(KDE): For each dimension, we fit a kernel density estimator (KDE) on $\mathcal{D}_{\text{diffusion}} = \{s_t^{(0)}\}_k$ and then we evaluate that probability density function for each point in $\mathcal{D}_{\text{timeseries}}$. Then, we take the product of these densities in each dimension to obtain the weight for each sample $\hat{s}_t^b$. We obtain a single representative sample by taking the weighted average of samples in $\mathcal{D}_{\text{timeseries}}$. To ensure stability,

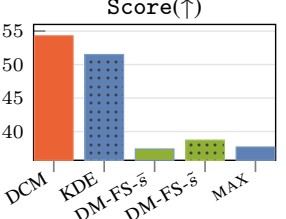

Figure 9: Candidate Selection

when the sum of the weights is near zero, we use the mean of the $\mathcal{D}_{\text{timeseries}}$ as the states. We use Scott's rule [24] to compute the bandwidth. DM-FS-$\bar{s}$, DM-FS-$\tilde{s}$ select the closest prediction from DM to the mean and median of the *Zero-Shot FM*'s predictions, respectively. FORL (MAX) constructs a diagonal multivariate distribution from the dimension-wise mean and standard deviation of the

Table 2: **Normalized scores (mean ± std.) for FORL and baselines on `maze2d-large`.** Bolds denote the best scores and those not significantly different (Welch's t-test, p > 0.05). Suffixes -T and -R denote the use of TD3+BC [2] and RORL [25], respectively.

| maze2d-large | TD3BC Policy | | | | RORL Policy | | | |
|---|---|---|---|---|---|---|---|---|
| | TD3BC | TD3BC+LAG-$\bar{s}$ | DMBP+LAG-T | FORL (ours)-T | RORL | RORL+LAG-$\bar{s}$ | DMBP+LAG-R | FORL (ours)-R |
| real-data-A | **14.7** ± 5.7 | 2.5 ± 2.7 | 4.8 ± 3.9 | **20.7** ± 3.5 | 12.2 ± 2.3 | 13.0 ± 2.0 | 4.3 ± 5.0 | **56.9** ± 3.0 |
| real-data-B | -0.9 ± 2.0 | 4.6 ± 8.9 | 11.7 ± 12.6 | **56.8** ± 14.4 | 1.2 ± 5.5 | 13.1 ± 14.7 | 28.5 ± 11.7 | **98.5** ± 19.0 |
| real-data-C | 0.8 ± 1.9 | 21.6 ± 8.4 | 29.5 ± 13.7 | **56.9** ± 14.6 | 3.1 ± 0.9 | 60.6 ± 8.5 | 39.4 ± 6.1 | **139.0** ± 15.1 |
| real-data-D | 2.5 ± 4.4 | 14.9 ± 4.3 | 14.4 ± 6.8 | **29.5** ± 10.3 | -1.6 ± 0.7 | 17.9 ± 6.8 | 17.5 ± 6.0 | **33.1** ± 2.3 |
| real-data-E | -2.3 ± 0.2 | 1.0 ± 4.2 | 2.0 ± 3.9 | **8.0** ± 4.2 | -0.9 ± 2.0 | 3.3 ± 4.4 | 2.2 ± 4.5 | **32.2** ± 15.3 |
| **Average** | 3.0 | 8.9 | 12.5 | **34.4** | 2.8 | 21.6 | 18.4 | **71.9** |

forecasted states, then selects the sample predicted by our diffusion model with the highest likelihood under that distribution. Although all baselines fuse information using the same two sets generated by the diffusion model and *Zero-Shot FM*, DCM has higher performance. In Table 8 we compute the maximum, minimum, and mean prediction error values over the test episodes used in Fig. 6. FORL (DCM) yields significantly stable prediction errors (`Maximum Error` ↓:**2.40**) for both maximum error and mean error compared to FORL (MAX) (`Maximum Error` ↓:**9.33**) demonstrating its robustness.

### 3.1.3 Intra-episode Non-stationarity

Our framework can natively handle intra-episode offsets, where the offset changes every $f = 50$ timesteps. In this setting, the offsets become available after the episode terminates, but the agent is subject to a time-dependent unknown offset within the episode. Zero-shot forecasting foundation module can generate samples before the episode begins. Our diffusion model (FORL-DM) itself does not rely on the forecasts of the foundation module and only tracks observation changes and actions which are invariant to the offsets. The DCM can adaptively fuse information from both models at each timestep without requiring any hyperparameters. Table 10 and Fig. 10 show the average scores for DQL vs. FORL-DM and

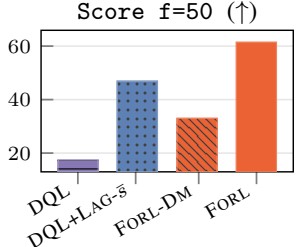

Figure 10: Intra-Episode Performance

DQL+LAG-$\bar{s}$ vs. FORL. Among the algorithms that do not use any past ground truth offsets DQL and FORL-DM, only using the diffusion model of FORL significantly increases performance. When we have access to past offsets, FORL obtains a superior performance. This shows that our method covers both cases, when information is available and not available, even when offsets are not constant throughout the episode.

### 3.1.4 Offset-Scaling

We scale the offsets with $\alpha$ across all maze experiments. We conduct experiments in 5 environments (all antmaze and maze2d used in Table 1) across 5 time-series dataset setups with $\alpha \in \{0, 0.25, 0.5, 0.75, 1.0\}$, where $\alpha = 0$ is the standard offline RL environment used during training and $\alpha = 1.0$ is our evaluation setup. The results show that FORL outperforms the baselines, confirming its robustness. Even a small scaling of $0.25$ results in a sudden drop in

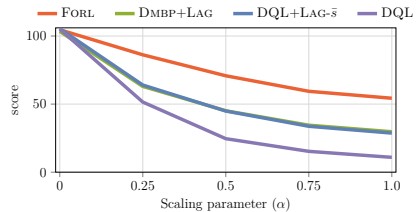

Figure 11: Impact of offset scaling ($\alpha$) on average normalized scores.

performance, whereas FORL only experiences a gradual decrease in Figure 11. Detailed results for each environment and $\alpha$ pairs are provided in Appendix Figure 13.

### 3.1.5 Policy-Agnostic

In the `maze2d-large` experiments (in Table 2, maze2d-medium in Appendix Table 3), we use Robust Offline RL (RORL) [25], and TD3BC [2] offline RL algorithms instead of DQL [14], to analyze the effect of offline RL policy choice during evaluation. RORL+LAG-$\bar{s}$ and TD3BC+LAG-$\bar{s}$ extend RORL and TD3BC by using the sample mean of the forecasted states $\{\hat{s}_t\}_l$ at each timestep (using the constant per-episode predicted $b^j$). Results indicate that using a robust offline RL algorithm during training significantly increases performance (71.9) compared to DQL (38.8) and TD3BC (34.4) at test time when used with FORL, no increase when used alone, and a marginal increase with Lag-Llama and DMBP+LAG. We observe similar performance gains when applying FORL to other

policies (Section E), including Implicit Q-Learning (IQL) [26] and FQL [23], as detailed in Table 4 and Table 7.

## 4 Related Work

**Reinforcement Learning in Non-Stationary Environments**   Existing works in non-stationary reinforcement learning (RL) predominantly focus on adapting to changing transition dynamics and reward functions. Ackermann et al. [27] propose an offline RL framework that incorporates structured non-stationarity in reward and transition functions by learning hidden task representations and predicting them at test time. Although our work also investigates the intersection of non-stationary environments and offline RL, we assume stationarity during training. To learn adaptive policies *online*, meta-learning algorithms have been proposed as a promising approach [28–30]. Al-Shedivat et al. [29] explores a competitive multi-agent environment where transition dynamics change. While these approaches provide valuable insights, they often require samples from the current environment and struggle in non-trivial non-stationarity, highlighting the need for more future-oriented methods [9, 31]. Examples of such future-oriented approaches include Proactively Synchronizing Tempo (ProST) [9] and Prognosticator [31], which address the evolution of transition and reward functions over time. ProST leverages a forecaster, namely, Auto-Regressive Integrated Moving Average (ARIMA), and a model predictor to optimize for future policies in environments to overcome the time-synchronization issue in time-elapsing MDPs. This approach aligns with our focus on time-varying environments and similarly utilizes real-world finance (e.g., stock price) time-series datasets to model non-stationarity. Both ProST and Prognosticator assume that states are fully observable during testing and that online interaction with the environment is possible during training—conditions that are not always feasible in the real world. Instead, our approach assumes that states are not fully observable and that direct interaction with the environment during training is not feasible, necessitating that the policy be learned exclusively from a pre-collected dataset.

**Robust offline RL**   Testing-time robust offline RL methods DMBP [3], RORL [25] examine scenarios where a noise-free, stationary dataset is used for training, but corruption is introduced during testing. This is distinct from [3], training-time robust offline RL [32, 33], which assumes a corrupted training dataset. Both RORL [25] and DMBP [3] assume access only to a clean, uncorrupted offline RL dataset, as FORL, and they are evaluated in a perturbed environment. To the best of our knowledge, FORL is the first work to extend this setting to a non-Markovian, time-evolving, non-stationary deployment environment. We focus on time-dependent exogenous factors from real-data that are aligned with the definition of a non-stationary environment [6].

**Diffusion models in offline RL**   Diffusion models [12] have seen widespread adoption in RL [34, 35] due to their remarkable expressiveness, particularly in representing multimodal distributions, scalability, and stable training properties. In the context of offline RL, diffusion models have been used for representing policies [14, 36–38], planners [39, 40], data synthesis [41, 42], and removing noise [3]. Notably, Diffusion Q-learning [14] leverages conditional diffusion model policies to learn from offline RL datasets, maintaining proximity to behavior policy while utilizing Q-value function guidance. In contrast, our method harnesses diffusion models to learn from a sequence of actions and effect tuples, leveraging the multimodal capabilities of diffusion models to identify diverse candidate locations of the hidden states.

## 5 Conclusion

We introduce **F**orecasting in Non-stationary **O**ffline **RL** (FORL), a novel framework designed to be robust to passive non-stationarities that arise at test time. This is crucial when an agent trained on an offline RL dataset is deployed in a non-stationary environment or when the environment begins to exhibit partial observability due to unknown, time-varying factors. FORL leverages diffusion probabilistic models and zero-shot time series foundation models to correct unknown offsets in observations, thereby enhancing the adaptability of learned policies. Our empirical results across diverse time-series datasets, OGBench [21] and D4RL [15] benchmarks, demonstrate that FORL not only bridges the gap between forecasting and non-stationary offline RL but also consistently outperforms the baselines. Our approach is currently limited by the assumption of additive perturbations. For future work, we plan to extend our work to more general observation transformations.

## Acknowledgments and Disclosure of Funding

Georg Martius is a member of the Machine Learning Cluster of Excellence, EXC number 2064/1 – Project number 390727645. Co-funded by the European Union (ERC, REAL-RL, 101045454). Views and opinions expressed are, however, those of the author(s) only and do not necessarily reflect those of the European Union or the European Research Council. Neither the European Union nor the granting authority can be held responsible for them. This work was supported by the German Federal Ministry of Education and Research (BMBF): Tübingen AI Center, FKZ: 01IS18039A. This work was in part supported by the INVERSE project (101136067) funded by the European Union and JSPS KAKENHI Grant Numbers JP23K24926, JP25H01236. The numerical calculations reported in this paper were partially performed at TUBITAK ULAKBIM, High Performance and Grid Computing Center (TRUBA resources). The authors would like to thank René Geist, Tomáš Daniš, Ji Shi, and Leonard Franz for their valuable comments on the manuscript.

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
