# OpenReview forum: "Forecasting in Offline Reinforcement Learning for Non-stationary Environments"
_NeurIPS.cc/2025/Conference — NeurIPS 2025 spotlight_

### Official Review · Reviewer_x1rM · 2025-06-30

**Clarity:** 2
**Significance:** 3
**Originality:** 3
**Rating:** 4
**Confidence:** 3

**Summary:**

This paper introduces Forecasting in Offline RL (FORL), a framework for enabling offline RL agents to handle a specific kind of non-stationarity at test time: unobserved observation offsets that vary between episodes but are constant within episodes. FORL uses a zero-shot time-series foundation model to sample candidate offsets for the current episode, assuming past ones are revealed after a delay. In parallel, it uses a conditional diffusion model to sample candidate Markov states given a within-episode sequence of actions and "delta observations" ($o_t-o_{t-1})$. The two sets of samples are combined via a "dimension-wise closest match" heuristic to give a final state estimate that is fed to the policy. The authors demonstrate FORL's effectiveness on standard offline RL benchmarks augmented with observation offsets derived from real-world time-series data, showing improved performance over existing methods.

**Questions:**

1. Clearly, the main limitation of this work is its technical assumptions: episode-constant offsets and revealing of ground truth offsets after a delay. Could you discuss the technical barriers that prevent the current FORL framework from handling continuously changing non-stationarity, and ideally bypassing the dependency on delayed ground truth offset information? What architectural changes would be necessary to tackle this more realistic and general problem?

2. Given the framework's dependency on an external forecaster, have you performed any analysis on the sensitivity of FORL to forecasting errors? How does performance degrade as the forecast becomes less accurate? This would provide important context for the method's robustness.

**Ethical Concerns:**

["NO or VERY MINOR ethics concerns only"]

**Final Justification:**

The authors' detailed reply has largely addressed my concerns about relying on episode-constant offsets and the visibility of ground truth offsets after a delay. These appeared to be major limitations, but don't in fact seem to be fundamental. This new information is sufficient for me to raise my score to a 4. I suggest that the authors spend time rewriting the paper to highlight this, to prevent future readers from having similar concerns.

**Limitations:**

Yes. Because the authors are very transparent about their restricted problem definition, I don't think a lack of discussion is an issue. However, as outlined above, I think this limitation *itself* is a major drawback of the paper and my primary reason for a lowered score.

**Paper Formatting Concerns:**

None.

**Quality:**

3

**Strengths And Weaknesses:**

Strengths:
- The paper does a good job of formally defining a specific and challenging sub-problem within the broader space of non-stationary RL. The scope of the work is clear throughout, which allows for a focused solution and rigorous evaluation.
- The core idea of combining a long-range, inter-episode forecaster with a short-range, intra-episode state inference model seems quite original. Using a conditional diffusion model for belief state generation based on action-delta trajectories is a technically innovative approach that is distinct from typical uses of diffusion models in RL.
- Within the confines of the specific problem being studied (episode-constant offsets), FORL significantly outperforms relevant baselines. The use of real-world time series data to generate the offsets makes the evaluation more compelling than using simple synthetic noise.

Weaknesses:
- The paper's biggest weakness is its exclusive focus on observation offsets that are constant within an episode. This is a significant constraint that does not capture most real-world non-stationarities (e.g. sensor drift, mechanical wear, environmental changes), which typically evolve continually during a task. The entire technical approach relies on this assumption and would likely fail dramatically if it was broken. This makes the current method a solution to a highly stylised and arguably artificial variant of the non-stationarity problem.
- Moreover, the zero-shot forecasting model makes a further assumption that ground truth episode offsets are revealed after a delay (of $P$ episodes). It is unclear why this assumption is reasonable, and it doesn't appear to be justified anywhere in the paper.
- The method's performance is heavily dependent on the quality of the pre-trained time-series foundation model. The paper does not analyse the sensitivity of FORL to forecasting errors or to environmental dynamics that are out-of-distribution for the forecaster, which is a crucial consideration for any real-world deployment.
- (Minor) The method's name "Forecasting in Offline RL" is not particularly informative as it doesn't include any reference to non-stationarity. The authors should consider a different name.

---

> ### Author Rebuttal · Authors · 2025-07-31
>
> We thank the reviewer for their inspiring and constructive feedback on our paper, and recognizing the **clarity** of our problem formulation, the novelty of our conditional diffusion-based approach, and our **rigorous evaluation**.
>
> >The paper's biggest weakness is its exclusive focus on observation offsets that are constant within an episode. This is a significant constraint that does not capture most real-world non-stationarities (e.g. sensor drift, mechanical wear, environmental changes), which typically evolve continually during a task. The entire technical approach relies on this assumption and would likely fail dramatically if it was broken. This makes the current method a solution to a highly stylised and arguably artificial variant of the non-stationarity problem.
>
> We sincerely thank the reviewer for raising this critical point, and highlighting its importance throughout their review. The reviewer’s insightful comment prompted us to conduct new experiments to empirically validate the generality of our framework. The challenge of intra-episode non-stationarity is indeed a crucial frontier for the field, and we are grateful for the opportunity to demonstrate our method's capabilities in this more general setting. We are pleased to report that it natively handles timestep-wise non-stationarities without requiring any modifications to the architecture or the framework. The results presented below confirm that our method remains highly effective with in-episode changes to observation offsets. We believe this demonstrates a significant step beyond prior work that relies on simpler parametric functions, as our method successfully adapts to challenging dynamics from real-world time-series data.
>
> **Performance on Timestep-Wise Additive Offsets**
>
> Zero-shot forecasting foundation module can continuously generate samples, our diffusion model itself does not rely on the the forecasts of the foundation module and only tracks observation changes and actions which are invariant to the offsets, our DCM module already samples a good estimate at each timestep without requiring any hyperparameters. This shows that our algorithm can solve intra-episode offsets, we will add this to our contributions.
>
> Below we present results for the case where the offsets change every $f$ timesteps inside the episode. We take $f=50$ for the experiments below. We observe that our framework can handle also the within episode change counterpart. It can also handle every timestep changing drifts, which we believe is a very important problem and fits not only into the sequential episodic scenario but also to the time-step evolving scenario, where the offsets become available after the episode terminates, but the agent is subject to a time-dependent non-markovian shifts within the episode. Episodic changes are usually addressed in the literature due to their tractability but we uncover that time-step wise changes is also possible which makes our framework foundational in terms of extendability.
>
> | Method             | Score   |
> |--------------------|--------:|
> | **FORL(ours)**    |**61.5** |
> | FORL-DM(ours)    | 33.0|
> | DQL(Mean+Lag)     |46.9    |
> | DQL            | 17.4 |
>
> Among the algorithms that do not use any past ground truth offsets $FORL-DM$ and DQL,  only using the diffusion model of FORL $FORL-DM$ significantly increases performance. Integration of the forecasting FM significant increase in performance, when we have access to this information FORL as a whole obtains a superior performance, this shows that our method covers both cases when information is available and not available even when there is intra-episode changes in the offsets.
>
> > Moreover, the zero-shot forecasting model makes a further assumption that ground truth episode offsets are revealed after a delay (of  episodes). It is unclear why this assumption is reasonable, and it doesn't appear to be justified anywhere in the paper.
>
> We thank the reviewer for this question, as it allows us to highlight a crucial aspect of our method. We would like to clarify that the assumption of receiving delayed ground truth offsets is **not required** for our method to outperform baselines.
>
> Our proposed diffusion module, even without access to any ground truth offsets, demonstrates superior performance in environments with continuously changing non-stationarities. This is empirically validated in our ablation studies presented in Supplementary Material F.3 (Figures 11). Given the importance of this finding, we will move these figures and the corresponding discussion to the main body of the paper to make this capability explicit. Below we share the numerical values of the plot in the main paper:
>
> | Method             | Score   |
> |--------------------|--------:|
> | $\texttt{DM}$ (renamed to FORL-DM)| **27.4** |
> | $\texttt{DQL}$                   | 10.9    |
> | $\texttt{Med-DCM}$               | 17.7     |
> | $\texttt{Med-Noise}$             | 15.5     |
> | $\texttt{H-Lag-DCM}$             | 26.3     |
> | $\texttt{H-Lag}$                 | 17.3     |
>
>
> > Clearly, the main limitation of this work is its technical assumptions: episode-constant offsets and revealing of ground truth offsets after a delay. Could you discuss the technical barriers that prevent the current FORL framework from handling continuously changing non-stationarity, and ideally bypassing the dependency on delayed ground truth offset information? What architectural changes would be necessary to tackle this more realistic and general problem?
> > I think this limitation itself is a major drawback of the paper and my primary reason for a lowered score.
>
> 1.  **Handling Continuously Changing Non-Stationarity:** As demonstrated in our first response, the FORL framework is not fundamentally limited to episode-constant offsets. The primary barrier is the predictive capability of the external forecaster. To handle more complex, continuously changing dynamics over long horizons, one simply needs a more powerful time-series foundation model. The architecture of FORL itself does not need to change; it is designed to integrate any such forecaster.
> 2.  **Bypassing Dependency on Ground Truth Offsets:** We wish to re-emphasize that **no architectural changes are necessary** to operate without delayed ground truth information. Our DM is inherently capable of functioning in a purely zero-shot manner, **without** leveraging the forecasts directly. As empirically demonstrated in our ablations (Figures 11 in the appendix), this configuration already outperforms existing baselines, and heuristics.
>
> >The method's performance is heavily dependent on the quality of the pre-trained time-series foundation model. The paper does not analyse the sensitivity of FORL to forecasting errors or to environmental dynamics that are out-of-distribution for the forecaster, which is a crucial consideration for any real-world deployment.
> >Given the framework's dependency on an external forecaster, have you performed any analysis on the sensitivity of FORL to forecasting errors? How does performance degrade as the forecast becomes less accurate? This would provide important context for the method's robustness.
>
> The time-series dataset plots with mean, ground truth values are available in the supplementary, we report results in each dataset so we can see the degradation in FORL and baselines
> While a full sensitivity analysis is implicitly present in our results (forecasting errors for each dataset in Table 1, for their corresponding plot in Fig. 13). Our experiments are conducted across five different real-world time-series datasets, each presenting a unique forecasting challenge. Table 3 shows the comparison of prediction errors, where the $DQL-Lag-\bar{s}$ uses just the forecasting module, which is detailed in Supplementary Material Section E. For instance for the real-data-E where the forecasts are highly inaccurate performance in all methods degrade, whereas FORL is comparatively more robust. We are currently working on alternative ways of analysis following reviewer's feedback.
>
> >  (Minor) The method's name "Forecasting in Offline RL" is not particularly informative as it doesn't include any reference to non-stationarity. The authors should consider a different name.
>
> We agree and will rename our method to either "Prospective Offline RL" or "Anticipatory Offline RL." We chose the original name to attract interest from the time-series forecasting community, as we believe time-series forecasting is highly relevant to the general problem of Non-stationary RL.

---

> > ### Author Response · Authors · 2025-08-04
> >
> > We thank the reviewer for their insightful question regarding the forecaster's accuracy. To analyze the sensitivity of **FORL** to forecasting errors, we have compared its performance against **DQL+MeanLag** that only uses the forecaster's predictions. The table below presents the average prediction error ($\downarrow$) across all datasets in antmaze-umaze/medium/large-diverse and maze2d-medium/large environments  with 5 seeds.
> >
> > | Dataset |**DQL+MeanLag** | **FORL** | *Error Reduction* |
> > | :--- | :---: | :---: | :---: |
> > | $\texttt{real-data-A}$ | 4.56 | **3.32** | 27.0% |
> > | $\texttt{real-data-B}$ | 3.66 | **2.29** | 37.4% |
> > | $\texttt{real-data-C}$ | 3.0  | **2.69** | 10.2% |
> > | $\texttt{real-data-D}$ | 4.29 | **1.87** | 56.5% |
> > | $\texttt{real-data-E}$ | 5.45 | **5.21** | 4.3% |
> >
> > In all datasets, **FORL** outperforms the **DQL+MeanLag** baseline. **FORL** achieves its greatest impact on moderately challenging forecasts (a 56.5% error reduction on $\texttt{real-data-D}$)). Its behavior at the extremes further demonstrates its robustness:
> >
> > - **FORL** still refines the best forecast by 10.2% ($\texttt{real-data-C}$)
> > - **FORL** improves the worst forecast by 4.3% ($\texttt{real-data-E}$).

---

> > > ### Author Response · Authors · 2025-08-04
> > >
> > > To ensure we address your question regarding *“analysis on the sensitivity of FORL to forecasting errors”* as precisely as possible, we would like to clarify our understanding. If the question is about performance across different datasets ($\texttt{real-data-A,B,C,D,E}$) with varying forecasting difficulty, our previous table analyzing prediction errors is the most relevant. If the question is about a sensitivity test to offset magnitude, we believe our **Offset Scaling ($\alpha$) Ablation Study in our main paper (&sect; 4.2, L362-L374, Fig. 8; Supplementary Fig.10)** directly addresses this. We would be grateful for your guidance on which analysis is aligned with your concern.

---

### Official Review · Reviewer_cynB · 2025-06-30

**Clarity:** 3
**Significance:** 2
**Originality:** 3
**Rating:** 4
**Confidence:** 4

**Summary:**

This paper proposes FORL, a framework for offline RL under test-time observation non-stationarity, where each episode is affected by a latent, fixed offset. The method uses a time-series model to forecast future offsets based on past episodes, and a diffusion model to correct state estimates during testing. A fixed offline-trained policy then operates on the corrected states. Experiments on D4RL benchmarks with synthetic offsets demonstrate performance gains over baseline methods in perturbed settings.

**Questions:**

1. If the test-time environment is a POMDP, why is the training data assumed to be from a stationary, fully observable MDP? Isn’t that an unrealistic assumption given the real-world motivation?
2. Why is the additive offset model (i.e., ot=st+bj) considered sufficient to represent the broad class of real-world observation non-stationarities?
3. How exactly is the forecasting model trained or used, given that ground-truth offset values are not observable during deployment? Where do these values come from?

**Ethical Concerns:**

["NO or VERY MINOR ethics concerns only"]

**Final Justification:**

The score is updated based on the rebuttal.

**Limitations:**

Yes

**Quality:**

2

**Strengths And Weaknesses:**

Strengths:
1. The paper focuses on a realistic and underexplored problem: test-time observation non-stationarity in offline RL. This is an important deployment issue that is often ignored by current benchmarks and methods.
2. The combination of diffusion-based state inference and time-series forecasting is novel and modular.

Weaknesses:
1. The paper assumes that the training data comes from a fully observable, stationary MDP, while the test-time deployment setting is a POMDP with unknown episode-wise observation offsets. If the real world is truly non-stationary (as the paper claims), then the offline dataset itself should already reflect these dynamics—i.e., it should be collected under a POMDP setting. The assumption that we have clean, unshifted ground-truth states during training but encounter sudden, unobservable bias at test time feels not well justified.
2. The forecasting component depends on a sequence of ground-truth offset values from past episodes but these offsets are fundamentally unobservable in practice. The model assumes they "become available" after each episode, which breaks the zero-shot narrative and introduces information leakage. In real deployment, these values are never accessible, making the forecasting model infeasible in practice.
3. Although the paper claims to tackle realistic non-stationarity, the experiments are all conducted on standard D4RL environments—clean MDPs—augmented by synthetic additive offsets on state vectors. These do not reflect real-world POMDP complexity like occlusion, information loss, non-linear emissions, or sensor dropout. The claim that this method handles realistic partial observability is not strongly supported.
4. The paper only compare against DQL and its variants, omitting widely-used and stronger baselines like IQL, or methods related to Decision Transformer.The modularity of the proposed method would allow easy integration with other offline RL methods, but this is not well tested.

---

> ### Author Rebuttal · Authors · 2025-07-31
>
> We thank the reviewer for their insightful and constructive feedback on our paper. We are encouraged that the reviewer recognizes the **novelty and modularity** of our approach and the importance of our problem setting. Following the reviewer's suggestions, we extend our empirical analysis, clarify our positioning and demonstrate the significance of our setup.
>
> > "The assumption that we have clean, unshifted ground-truth states during training but encounter sudden, unobservable bias at test time feels not well justified."
>
> We will revise our related work to reflect how our problem setting fits into the existing research on robust offline RL. Our choice of a stationary training dataset and a non-stationary test environment is a deliberate one that positions our work as a significant extension of the important and well-established subfield of (i) **testing-time robust offline RL**[RORL,DMBP]. This is distinct from, (ii) **training-time robust offline RL** frameworks [1,2], which assume a corrupted training dataset. The "clean train, perturbed test" is standard practice to evaluate testing-time robustnes. Both RORL and DMBP used in our baselines assume only having access to a clean uncorrupted offline RL dataset as FORL, and they are then evaluated on a perturbed environment. RORL can be integrated with DMBP and FORL which we provide results for both cases in Table 2 & 4 (L1099). To the best of our knowledge, **FORL is the first work to extend testing-time robust offline RL to a non-Markovian, time-evolving, non-stationary deployment environment.** We focus on time-dependent exogenous factors from real-data which is aligned with the non-stationary environment definition.
>
> > "If the test-time environment is a POMDP, why is the training data assumed to be from a stationary, fully observable MDP? Isn’t that an unrealistic assumption given the real-world motivation?"
>
> Our problem formulation, which utilizes a stationary training dataset and a non-stationary test environment, is motivated by scenarios where data collection and deployment occur under different conditions. Offline datasets are often collected during periods of stationary conditions, since the goal is not to collect the dataset again and only rely on a static pre-collected dataset. Hence, data may be collected using calibrated hardware, in controlled laboratory settings, or from validated simulators. However long-term deployment can expose the agent to systematic shifts, which were not present in the training data. Furthermore we may not have access to all possible offsets evolution patterns that can occur during training, which is the focus of **testing-time robust offline RL**. Our setup enables the agent to learn the core task dynamics from the clean offline data while treating the non-stationarity as a separate problem to be solved zero-shot at test time, decoupling policy and dynamics learning from the challenge of adapting to unseen non-stationary patterns during deployment.
>
> The key advantage of our testing-time adaptation approach is its **generalization to unseen non-stationarity**. An approach trained on a dataset that already contains specific non-stationary patterns would learn to model *those specific patterns*. If a new, previously unseen pattern of non-stationarity emerges at test-time, such a model would can fail, necessitating a costly cycle of **new data collection, retraining, and hyperparameter tuning**.
>
> FORL, by contrast, is designed to be robust to this scenario, which aligns with the general aim of offline RL where additional data collection is restricted. It learns the time-invariant dynamics of the task from the clean, stationary data. The adaptation to a different non-stationary pattern is then handled at test-time by its modular components. This makes our framework significantly more flexible and practical for real-world deployment, where the future evolution of the environment can be fundamentally unknown. We will clarify this crucial distinction in the related work section.
>
> [1]Ye, Chenlu, Rui Yang, Quanquan Gu, and Tong Zhang. "Corruption-robust offline reinforcement learning with general function approximation." NeurIPS 2023
>
> [2]Zhang, X., Chen, Y., Zhu, X. and Sun, W., Corruption-robust offline reinforcement learning. AISTATS 2022
>
> > Why is the additive offset model (i.e., ot=st+bj) considered sufficient to represent the broad class of real-world observation non-stationarities? & "episode-wise observation-offsets"
>
> We do not claim that the additive offset model is universally sufficient for all possible real-world non-stationarities. Instead, we propose this formulation as a critical and more generalizable first step beyond the restrictive assumptions in prior literature. Existing work models non-stationarity using predefined, often simplistic, functional forms. A common approach is to use a sinusoidal function where the episode index is the input, which implicitly assumes a long-term, repetitive pattern in the environmental changes. Our work makes no such assumption of repetition. Our framework, by contrast, conceptualizes the non-stationarity as a time-series component. Although we agree that $o_t=s_t+b_j$ is general, the offset term $b_j$ can be **large**, complex, and non-Markovian as illustrated in Fig. 3 Fig.6. We position our work as a foundational framework that is designed to scale with rapid advancements in sequential modeling. As the capabilities of foundation models improve, enabling longer-horizon predictions, the class of non-stationarities our method can effectively handle will expand accordingly.
>
> >  The forecasting component depends on a sequence of ground-truth offset values from past episodes
>
> We thank the reviewer for raising this critical point, and we apologize if our description was unclear. We will revise the text to be more precise.
>
> The assumption is **not that offsets become available after every episode, but after a long interval of P episodes** (e.g., P=60 in our experiments). This models scenarios like periodic maintenance checks on a robot or quarterly reconciliation of financial data, where a ground-truth signal becomes available, but only infrequently. In many deployments offsets are logged after scheduled calibration (e.g., daily robot zeroing, weekly clinical lab alignment). Hence they are **not revealed after every episode** and there is no leakage.
>
> Our framework can handle intra-episode offsets, where the offset changes every 50 timesteps, but the offsets are revealed after each episode. We share our results for this setting below across all antmaze, maze2d environments and real-data-A,B,C,D,E with 5 random seeds.
>
> | Method| Score|
> |--------------------|--------:|
> | **FORL(ours)**|**61.5** |
> | FORL-DM(ours)| 33.0|
> | DQL(Mean+Lag)|46.9|
> | DQL | 17.4 |
>
> Our framework's core contribution is **robust even when offsets are never revealed**. DM component, proposed as part of our FORL framework,generates candidate states from the sequence of action‑$\Delta_o$, does not require access to historical offset information. As shown in our results in Figure 11 and Section F.3 (L1121), the DM (renamed to FORL-DM) module by itself outperforms DQL baseline. We will move the DM and H-LAG-DCM to the main text to make it more visible.
>
> | Method| Score|
> |--------------------|--------:|
> | $\texttt{DM}$ (FORL)| **27.4** |
> | $\texttt{DQL}$| 10.9|
> | $\texttt{Med-DCM}$  | 17.7|
> | $\texttt{Med-Noise}$| 15.5|
> | $\texttt{H-Lag-DCM}$| 26.3|
> | $\texttt{H-Lag}$| 17.3|
>
> This demonstrates that our primary innovation—the within-episode state estimation based on action-effect history—is highly effective as a standalone module for the most challenging scenario where ground-truth offsets are never accessible. The full FORL framework (**54.25**), including the forecaster, is presented as an enhanced version for settings where infrequent, periodic ground-truthing is feasible. We thus provide a practical solution for both cases. Our "zero-shot" claim refers to the fact that the agent requires no training on non-stationary data and can adapt to unseen non-stationary patterns from the first timestep. We will clarify this distinction between the DM and full FORL settings in the paper.
>
> > How exactly is the forecasting model trained or used, given that ground-truth offset values are not observable during deployment? Where do these values come from?
>
> The forecaster is a pre‑trained Lag‑Llama foundation model. We use the open-source pretrained model and predict the future P offsets zero-shot from the same model for all data, based on only the context length of ground truth offsets. We will make this more explicit in the main text.
>
> > The paper only compare against DQL and its variants, omitting widely-used and stronger baselines like IQL, or methods related to Decision Transformer.The modularity of the proposed method would allow easy integration with other offline RL methods, but this is not well tested.
>
> We compared with DQL, RORL and TD3BC in our main paper. Results in DQL paper show that DQL achieves performance superior to IQL, and DT across D4RL benchmarks.
>
> DT: Antmaze and maze2d datasets require stitching, hence it is suggested in [3] to use Q-learning based methods instead. We did consider DT as a baseline, as it is a widely used and strong offline RL baseline, however our results ranging from (-4.02 to 0) confirm the near zero scores reported in [3], hence we omit DT at this time. Results below are across all real-data-A,B,C,D,E, and 5 seeds for environments maze2d-medium,large, antmaze-umaze and medium for IQL.
>
> |  | $\mathbf{IQL}$ | $\mathbf{IQL+MeanLag}$ | $\mathbf{DMBP+Lag-I}$ | $\mathbf{FORL-I}$ |
> |:---|:---:|:---:|:---:|:---:|
> | Score | 9.1 | 22.7 | 23.3 | **35.9** |
>
> [3]Bhargava, P., Chitnis, R., Geramifard, A., Sodhani, S. and Zhang, A., 2023. When should we prefer decision transformers for offline reinforcement learning?.ICLR,2024

---

> > ### Comment · Reviewer_cynB · 2025-08-02
> >
> > Thanks for the response! The authors have addressed all my concerns. I will increase my rating to 4.

---

> > > ### Author Response · Authors · 2025-08-04
> > >
> > > Thank you for your great suggestions and increasing your rating! We are grateful that we have addressed your concerns, please let us know if you have any further questions.

---

### Official Review · Reviewer_VqtP · 2025-07-01

**Clarity:** 2
**Significance:** 3
**Originality:** 3
**Rating:** 5
**Confidence:** 3

**Summary:**

This paper focuses on robust offline reinforcement learning, in the sense that at training time, the agent will receive clean, fully observable data, while at test time, the agent's data can be offset by certain (possibly non-Markovian) factors in the environment, rendering it partially observable. The proposed framework, FORL, plans using a diffusion model and diffusion policy learned over clean, offline trajectory data, and allows for forecasting without certain assumptions being made on either the nature of perturbations at test time, and handles nonstationarities without the need of explicitly modeling this problem as a POMDP.

**Questions:**

I think that there are some things to clean up in general about the paper -- I can understand how sampling is done with this model, but there is no Equation (6) for instance that is mentioned in Algorithm 1 on page 5. I am also not entirely sure what this DCM procedure is -- it seems like some nearest neighbor-style thing to generate good samples, but I don't understand what the unimodal sample dataset does here (because as far as I can understand, this unimodal dataset contains offsets from the time-series FM, not true samples). Can this be cleared up a bit? It's also not clear what the initial samples from the FM do in FORL in Algorithm 1 -- it seems like they just aren't used in the algorithm at all?

**Ethical Concerns:**

["NO or VERY MINOR ethics concerns only"]

**Final Justification:**

Given the OGBench results provided by the authors, I am happy to raise my score. Sorry this is coming so late -- I was at a conference last week.

**Limitations:**

Not explicitly, but it may not be needed.

**Paper Formatting Concerns:**

None for me.

**Quality:**

3

**Strengths And Weaknesses:**

The paper is reasonably clear -- I don't have a background in this specific area of offline RL, but I could understand the overall method after a couple of passes through it. The paper's motivation is also quite straightforward -- if it is possible to do forecasting (e.g. with the time series foundation model) to estimate the offsets, while also having an expressive generative model (e.g. the diffusion model) model the transition dynamics, the hope is that you can generalize well out of distribution by doing online planning by generating perturbed trajectories from the combination of these two models. Experimental results look pretty good too across a diverse suite of MuJoCo tasks common for offline RL.

I think from an empirical standpoint, the one weakness I could see is that D4RL is a somewhat saturated set of domains -- there are new, difficult tasks that are also fully-observable (such as OGBench) that would be great to see results for.

---

> ### Author Rebuttal · Authors · 2025-07-31
>
> We sincerely thank the reviewer for their valuable suggestions and encouraging feedback on our work. We are glad they found the paper's motivation straightforward, the experimental results strong. The reviewer's thoughtful questions have highlighted several areas where we can significantly improve the clarity and empirical scope of our paper.
>
> > I think from an empirical standpoint, the one weakness I could see is that D4RL is a somewhat saturated set of domains -- there are new, difficult tasks that are also fully-observable (such as OGBench) that would be great to see results for.
>
> OGbench is an excellent recent benchmark that was primarily developed for goal-conditioned offline RL but also includes standard offline RL experiments. Thank you for suggesting it. The current paper's scope was an extensive study to establish the method's core properties across different non-stationarities.
> Most testing-time robust offline RL algorithms use D4RL (RORL,DMBP), so hyperparameter optimization at this stage may not be possible for all methods, as the navigation environments are brittle against hyperparameters. However, we believe our empirical results remain valid since all baselines and testing-time robust offline RL algorithms use these established hyperparameters on D4RL tasks.
>
> > I think that there are some things to clean up in general about the paper -- I can understand how sampling is done with this model, but there is no Equation (6) for instance that is mentioned in Algorithm 1 on page 5.
>
> Thank you for your suggestion. We agree moving Eq.6 from the Supplementary Material (C.1) to the main text is important for the clarity of our paper. Eq.6 show the reverse diffusion chain and how it is formulated to generate the candidates states conditioned on $\tau_{t,w}$. The final samples of this reverse diffusion chain $s^0$ form our multimodal set from the diffusion model $\{s_{t+1}^{(0)}\}_k$.
>
> > I am also not entirely sure what this DCM procedure is -- it seems like some nearest neighbor-style thing to generate good samples, but I don't understand what the unimodal sample dataset does here (because as far as I can understand, this unimodal dataset contains offsets from the time-series FM, not true samples). Can this be cleared up a bit?
>
>
> We thank the reviewer for giving us the opportunity to clarify the datasets used in our `Dimension-Wise Closest Match` (DCM) procedure. The reviewer's understanding of DCM is **correct**. The unimodal distribution in here is the one predicted by the time-series FM. We are able to obtain samples from it by querying the model. We confirm and clarify below:
> - DCM is a nearest neighbor-style, parameter-free, data-driven approach method to generate a sample using information from a multimodal and a unimodal distribution.
> - Unimodal dataset does not contain true samples, as correctly pointed out by the reviewer. The unimodal dataset contains the offset removed states, where the offsets are predicted by the forecasting module. Time-series FM is a probabilistic forecasting, foundation model thus it proposes $l$ number of offsets for each episode. These offsets $\{\hat{b}\}_l$ are then subtracted from the observation $o_t$. The resulting offset removed states $\{\hat{s}^b_t\}_l$ forms the set of plausable states hypothesized by the forecasting module. The "unimodal" dataset contains forecasted offset removed states. We refer to the lines 235-L236 in our current paper below:
>
> "The *Zero-Shot FM* generates $l$ samples of offsets $\{\hat{b}\}_l$ from which we compute $\{\hat{s}^b_t\}_l = o_t - \{\hat{b}\}_l$."
>
> We will include this part explicitly in Algorithm 1 to be clearer. Furthermore, we will rename the Datasets useed in the DCM to explicitly refer which dataset it refers to instead of using *D*_(uni) and *D*_(multi).
>
> To summarize, as our diffusion model is learning a multimodal distribution, the candidate states generated by the diffusion model are multimodal (*D*_(diffusion), illustrated as filled orange circles in Fig.3), while the zero-shot time-series FM is unimodal (*D*_(timeseries)). We will denote *D*_(uni) by *D*_(timeseries) and *D*_(multi) by *D*_(diffusion).
>
> > It's also not clear what the initial samples from the FM do in FORL in Algorithm 1 -- it seems like they just aren't used in the algorithm at all?
>
> We are very grateful to the reviewer for their detailed reading and for spotting a crucial typo in Algorithm 1. The variables should be $\hat{b}$ (for bias/offset), not $\hat{d}$. These forecasted offsets $\{\hat{b}\}^p_l$ for the current episode $p$ are essential and used in lines Alg. 5,14, and 16. We correct this typo below, where the first line of the algorithm is changed to:
>
> Sample $\{\{\hat{b}\}^1_l, \dots, \{\hat{b}\}^P_l\}\sim $ *Zero-Shot FM*
>
> To obtain offset removed states from the forecaster we compute $\tilde{s}\leftarrow o_0-\{\hat{b}\}^p_l$ where $p$ is the episode, and $l$ is the number of forecasts samples per episode. Because we use a probabilistic forecasting module, the FM gives us multiple samples for that specific episode.
>
> We use initial samples from the FM in lines Alg. 5,14, and 16 where we update $\tilde{s}\leftarrow o_0-\{\hat{b}\}^p_l$ where p is the episode. We will make it clearer in the algorithm by adding $\tilde{s}\leftarrow o_0-\{\hat{b}\}^p_l$ after Line 4, Line 11 in the algorithm to explicitly show how the Samples from Zero-Shot FM are used to obtain the states from the forecasting module.

---

> > ### Comment · Reviewer_VqtP · 2025-08-04
> >
> > Hi,
> >
> > Thank you for all of your clarifications! This has been very helpful for me.
> >
> > In terms of the OGBench benchmarking, thank you for acknowledging this. As far as I know, hyperparameter optimization may not be central to OGBench (in fact, the original OGBench hparams for their benchmarking results use pretty much the same hparams for the corresponding algorithms in D4RL, so maybe minimal effort is required for that).
> >
> > Given the need to clear up a few things and fix up the paper structure a bit, I will not be changing my score at the moment. However, I do think this paper has a lot of potential and has some interesting, novel results, which has been shown in my initial score.

---

> > > ### Author Response · Authors · 2025-08-09
> > > **OGBENCH Results with Flow Q-Learning**
> > >
> > > We thank the reviewer for their very constructive suggestion on using the most recent **OGBench** [1]. We report results using **Flow Q-Learning (FQL)**[2], a recent offline RL algorithm that has achieved strong performance on OGBench. We use the hyperparameters suggested in the **Flow Q-Learning (FQL)**[2] open-source repository. In our experiments, FQL is evaluated in the same non-stationary setting as DQL, TD3BC, and RORL, as well as IQL.  For each task, we use the default $\texttt{singletask-v0}$ (\*). For the $\texttt{cube-single-play}$ manipulation experiments, we add offsets to the first 17 state dimensions, where each offset is drawn from a different time series in the corresponding GluonTS dataset. For each dataset  $\texttt{real-data-A,B,C,D,E}$, we specifically use its first 17 time series in GluonTS. The affected dimensions cover all joint positions, joint velocities, and end effector variables (position and yaw). Overall, the results show that $\mathbf{FORL-F}$ outperforms the baselines (including strong performance on manipulation tasks), indicating the robustness of our framework. The results in $\texttt{antmaze-large-navigate}$ is similar to the scores in our Table 1. where $\mathbf{DQL}$ was used. $\mathbf{FORL-F}$ uses the identical $\mathbf{FQL}$ policy, in line with our experimental setup throughout the paper.
> > >
> > > **Normalized scores (mean±std) for $\mathbf{FORL-F}$ and baselines on  $\texttt{antmaze-large-navigate-singletask-v0}$ (\*) and  $\texttt{cube-single-play-singletask-v0}$ (\*) tasks across 5 random seeds.**
> > >
> > > | Dataset | Time Series Dataset | $\mathbf{FQL}$ | $\mathbf{FQL+Lag-\bar{s}}$ | $\mathbf{FORL-F}$ |
> > > | :--- | :--- | :---: | :---: | :---: |
> > > | $\texttt{antmaze-large-navigate}$ | $\texttt{real-data-A}$ | $22.7 ± 2.2$ | $1.3 ± 0.7$ | $\textbf{24.3 ± 4.3}$ |
> > > |$\texttt{antmaze-large-navigate}$ | $\texttt{real-data-B}$  | $21.7 ± 5.4$ | $29.2 ± 8.8$ | $\textbf{40.0 ± 7.6}$ |
> > > |$\texttt{antmaze-large-navigate}$| $\texttt{real-data-C}$  | $5.0 ± 1.1$ | $34.6 ± 6.7$ | $\textbf{55.8 ± 3.7}$ |
> > > |$\texttt{antmaze-large-navigate}$ |  $\texttt{real-data-D}$  | $0.8 ± 1.9$ | $37.5 ± 5.1$ | $\textbf{75.8 ± 5.4}$ |
> > > | $\texttt{antmaze-large-navigate}$ | $\texttt{real-data-E}$ | $10.0 ± 4.1$ | $3.3 ± 0.0$ | $\textbf{15.3 ± 8.7}$ |
> > > | $\texttt{antmaze-large-navigate}$| **Average** | $12.0$ | $21.2 $ | $\textbf{42.2}$ |
> > > | -| - | - | - | - |
> > > |  $\texttt{cube-single-play}$  | $\texttt{real-data-A}$ | $0.0 ± 0.0$ | $0.0 ± 0.0$ | $\textbf{23.7 ± 3.6}$ |
> > > | $\texttt{cube-single-play}$  | $\texttt{real-data-B}$  | $0.0 ± 0.0$ | $15.0 ± 7.0$ | $\textbf{60.0 ± 7.0}$ |
> > > | $\texttt{cube-single-play}$  | $\texttt{real-data-C}$  | $0.4 ± 0.9$ | $10.0 ± 1.7$ | $\textbf{42.1 ± 5.6}$ |
> > > |  $\texttt{cube-single-play}$  | $\texttt{real-data-D}$  | $0.0 ± 0.0$ | $0.8 ± 1.9$ | $\textbf{70.0 ± 13.0}$ |
> > > | $\texttt{cube-single-play}$ | $\texttt{real-data-E}$ | $0.0 ± 0.0$ | $0.0 ± 0.0$ | $\textbf{32.7 ± 9.5}$ |
> > > |$\texttt{cube-single-play}$ | **Average** | $0.1$ | $5.2 $ | $\textbf{45.7}$ |
> > >
> > >
> > >
> > > [1]Park, Seohong, Kevin Frans, Benjamin Eysenbach, and Sergey Levine. "Ogbench: Benchmarking offline goal-conditioned rl." ICLR 2025
> > >
> > > [2]Park, Seohong, Qiyang Li, and Sergey Levine. "Flow q-learning." ICML 2025

---

> ### Author Response · Authors · 2025-08-04
>
> Thank you for your insightful and hopeful feedback. We are encouraged by your recognition of our work's potential and novel results! We are currently prioritizing OGBench integration.

---

> ### Author Response · Authors · 2025-08-09
>
> **Clarity and Paper Structure**
>
> We have already incorporated the requested clarifications, following the reviewer's suggestions. We also moved Equation 6 from Appendix $\S$ C.1 to the main text ($\S$ 2.2, L201). Below we detail the specific changes:
>
>
> *Clarifications*
>
> - L239 and L242. We renamed the lines in Fig. 6, and added $(D_{timeseries}$=$\big(\{\hat{s_{t}}\}^{b}\big)_{l})$ in L239.
>     - L239: $D_{uni}$ -> $D_{timeseries}$,
>     - L239, 242: $D_{multi}$ -> $D_{diffusion}$
>
> - L210: $\tilde{s}\leftarrow o_{0}-\overline{\big(\{\hat{b}\}^{p}_{l}\big)}$
>
> - L214:
>
> $\big(\{\hat{s_{t+1}}\}^{b}\big)_{l}$
>
> $\leftarrow o_{t+1}-\big(\{\hat{b}\}^{p}_{l}\big)$
>
> - L222: $\tilde{s}\leftarrow o_{t+1}-\overline{\big(\{\hat{b}\}^{p}_{l}\big)}$
>
> - L236: The $\textit{Zero-Shot FM}$ generates $l$ samples of offsets  $\big(\{\hat{b}\}^{p}_{l}\big)$ from which we compute **forecasted states** using
>
> $\big(\{\hat{s_{t}}\}^{b}\big)_{l}$
>
>  $=o_{t}-\big(\{\hat{b}\}^{p}_{l}\big)$.
>
> *Typographical error*
> - We fixed this from d (drift) ->b (bias) in L206
>
> We will also include all experiments conducted using OGBench and FQL. We thank the reviewer for their very constructive suggestions, and engagement with us during the rebuttal.

---

### Official Review · Reviewer_VWZ4 · 2025-07-02

**Clarity:** 2
**Significance:** 2
**Originality:** 3
**Rating:** 5
**Confidence:** 4

**Summary:**

This paper addresses a nonstationary offline RL setting in which the the offline dataset contains no nonstationarity, but during deployment there is an unknown, additive offset added to the state.
This offset evolves between episodes but is constant within each episode, which is also known as a Dynamic-Parameter MDP setting.
The authors propse to address the problem of identifying the offset by 1) a learned environment model (based on diffusion) and 2) a zero-shot time-series model, that uses the offsets from prior episodes.
In experiments on maze, antmaze, and in a manipulation task, the authors show that their method performs favorably compared to baselines, both in terms of achieved reward as well as prediction error.

**Questions:**

* Why can't the proposed method be extended to other transformations of the state? It seems that at least arbitrary monotonous transformations of the state should be within the scope of the proposed method?
 * Currently the knowledge of the offsets of past episodes is considered known. Is it possible to replace this with the output of the diffusion model for prior episodes? This would significantly increase the applicability of the proposed method.
 * Could you explain the decision to use "Dimension-Wise Closest Match" in your approach? There are many different ways to combine multiple different state estimators, so this choice currently feels a bit arbitrary.

Minor comments:
 * Equation (5) and (6) are referred to in the main text but only appear in the appendix.
 * L108 "FORL makes informed decisions at the start of each episode" does not seem like a contribution
 * L240: Maybe use "diffusion" and "timeseries" instead of "multimodal" and "unimodal"
 * I would recommend more clearly separating your proposed method from background material, such as the explanation of diffusion models.

**Ethical Concerns:**

["NO or VERY MINOR ethics concerns only"]

**Final Justification:**

The rebuttal has mostly resolved my concerns about the choice of the "Dimensionwise Closest Match" mechanism, by providing additional alternative baseline methods.

It has also fully resolved my concerns about the assumption of knowledge of the true offsets of past episodes, by showing that even without this assumption the method is useful.

I do have some remaining concerns about the clarity of the writing, but with some cleanup based on the rebuttals it should not be a major issue.

As such, I am raising my score 4->5 and recommending acceptance.

**Limitations:**

The assumption of the past offsets being revealed after the first few episodes seems strong. It would be helpful to evaluate the method in cases where it is violated.

**Quality:**

3

**Strengths And Weaknesses:**

Strengths:
 * The proposed problem setting of additive offsets during deployment with none being present during training is well motivated and challenging. Dynamics that differ during deployment are explored thoroughly in the robust RL literature and unknown transformations of the true state are throughly explored in the POMDP literature. However, the inclusion of unknown but predictably evolving transformations /offsets of the state is novel, to my knowledge.
 * The experimental validation shows a clear advantage of the proposed method

Neutral:
 * the application of zero-shot prediction models is relatively straight-forward. On the other hand, it is also very reasonable and it is unclear what else could be done in this setting.

Weakness:
 * the combination of the learned dynamics model and time-series model seems relatively ad-hoc. In essence, the proposed method obtains multiple estimates of the state from the dynamics model and chooses the closest pair among both sets, for each dimension. Thus the final estimate may have each dimension from a different sample drawn from the dynamics model, such that the combination could be entirely outside the support of the dynamics model. It is not clear why this method was used instead of any other way to select the state estimate. This is an important part of the algorithm and should be justified better.

 * The problem setting is limited to additive offsets, which is not a huge issue, but it is not clear to me why the method can not be extended to any other state transformations

* Experiments: Hyperparameter tuning is limited. While some hyperparameters of DMBP were tuned, for other baselines it seems like the default hyperparameters were used without tuning, which makes the comparison somewhat unfair as the proposed problem is new. Further, it would be helpful to evaluate more realistic cases of additive offsets, outside of maze-like. The method is also evaluated on a manipulation task, but there the advantage compared to baselines are less clear.


* The current method assumes knowledge of the true offset in past episodes, after P episodes have passed. This is a significant limitation and should be addressed in the paper, but is currently not listed in the limitations.

---

> ### Author Rebuttal · Authors · 2025-07-31
>
> We sincerely thank the reviewer for their inspiring and detailed comments. Your recognition of our **novel**, **well-motivated** and **challenging** problem setting and the experimental advantages of our method is truly encouraging. Your insightful questions and suggestions have highlighted key opportunities to strengthen our paper's contributions and structure.
>
> > Could you explain the decision to use "Dimension-Wise Closest Match" in your approach?.. currently feels a bit arbitrary.
>
> Thank you for this insightful question, and attention to detail. Although our choice of DCM was a principled one tailored to zero-shot adaptation, we agree it deserves a more thorough justification and analysis.
>
> The primary advantages of DCM are that it is simple, yet effective (Fig. 4), data-driven and non-parametric. In a test-time adaptation setting, where an agent may face entirely new non-stationary dynamics, methods that require tuning (e.g., the bandwidth of a Kernel Density Estimator) can be impractical. DCM provides a robust, data-driven way to fuse information from the two models without any tuning. It is designed for our setting where each state dimension can be affected by an independent non-stationarity to test the robustness of our framework, making a dimension-wise fusion a natural choice. We believe DCM can contribute to the research of fusion from different sources of information.
>
> To justify this choice empirically, we have conducted a new, extensive ablation study comparing DCM against five alternative fusion strategies (across 5 random seeds), including a Bayesian approach using KDEs (Bayes (KDE)) and methods that explicitly **ensure the final estimate lies within the support of the diffusion model's samples** (MAX(Appendix F.4), DM-mean(x)_(timeseries), DM-med(x)_(timeseries)). Maximum-likelihood approach was discussed in the supplementary material F.4 & F.5 (L1144), we include the numerical results for your convenience.
>
> - **Bayes KDE**: For each dimension, we fit a kernel density estimator (KDE) on D_(diffusion)=$\{\boldsymbol{s}t^{(0)}\}k$ and then we evaluate that probability density function for each point in D_(timeseries) Then, we multiply these densities to obtain the weight for each sample $\hat{s}^b_t$. We obtain a single representative sample from these distributions for taking the weighted average of samples in D_(timeseries)
> - **DM-mean(x)_(timeseries), DM-med(x)_(timeseries)** where we take the closest prediction from DM to the mean/median of the FM's predictions.
>
> | Method | Score|
> | :--- | :--- |
> | DCM (ours) | **54.3** |
> | Bayes (KDE) | 51.5 |
> | DM-mean(x)_(timeseries) | 37.4 |
> | DM-med(x)_(timeseries) | 38.7 |
> | MAX | 37.67 |
>
> The results show that our simple, hyperparameter-free DCM is highly competitive with, and often superior to, other methods. While the KDE-based `Bayes` method performs well, it requires bandwidth selection, and fallback mechanism for numerically unstable results. This highlights DCM's practical advantage.
>
> We will add a new subsection to the appendix detailing these alternative fusion methods and presenting the full comparison table. We will summarize these findings in the main text to provide a clear and compelling justification for our choice of DCM
>
> > Why can't the proposed method be extended to other transformations of the state?
>
> Our diffusion model component uniquely uses the $\Delta s, a$ pairs during training so  $\Delta s, a$ =  $\Delta o, a$ holds at test time.  Thus, other transformations are challenging.
> Nevertheless, we empirically evaluate our framework for scaling and offset transformations. Following the reviewer's suggestion, we conducted experiments with scaling using another time-series from the existing real-data domains. Our results for scaling and offsets in maze2d-large environment across all real-data-A,B,C,D,E are provided below:
>
> |  | $\mathbf{DQL}$ | $\mathbf{DQL+MeanLag}$ | $\mathbf{FORL}$ |
> |:---|:---:|:---:|:---:|
> | Score | 1.8 | 7.7 | **34.0** |
>
> > Hyperparameter tuning is limited.
>
> The underlying policies (TD3+BC, RORL, Diffusion-QL), and the forecasts predicted by the Zero-Shot FM are **identical policy checkpoints** for both our method and the baselines. We directly deploy the same policies without any modification in the policy modules of DMBP and FORL framework in Fig.2 as both have plug-and-play policy module. Therefore, any tuning of the policy would benefit all methods. Our focus on a zero-shot, test-time adaptation setting follows the testing-time robust offline RL setup. We will clarify this in the experimental setup section.
>
>
> > Further, it would be helpful to evaluate more realistic cases of additive offsets, outside of maze-like.
>
> We agree that testing on more realistic offsets is an important direction, which we intend to explore for future work.
>
> > Currently the knowledge of the offsets of past episodes is considered known.
>
> We apologize for not making this clearer in the main text: **our core contribution is highly effective even when we have no access to true offsets in past episodes.** The full FORL framework, which uses the forecasting foundation module, is designed for scenarios where this information is available. However, our use of a conditional diffusion model (DM) for candidate state generation conditioned on action‑$\Delta o$ trajectories trajectories offers a novel contribution distinct from typical diffusion model applications in reinforcement learning. DM can operate **without any access to historical offset information**, we wil rename DM as FORL-DM to be more precise. Specifically Fig.11 show that our Diffusion Module (DM) alone, without any historical offsets, outperforms all baselines that do not use historical ground-truth offsets.
>
> > Is it possible to replace this with the output of the diffusion model for prior episodes? This would significantly increase the applicability of the proposed method.
>
> We investigated this exact scenario with our H‑LAG+DCM, (L1140) (We correct the typo here where L1140 refers to H-LAG+DCM and 1136 refers to H-LAG where results for both algorithms are shown in Fig. 11). H‑LAG+DCM replaces true offsets with DM‑generated estimates and performs similarly to DM. This directly addresses the question of replacing known offsets with DM outputs. Our findings show comparable performance to the standalone DM.
>
> We will use a different coloring for methods that do not use past historical ground truth offsets (DQL, DM, MED-DCM, MED+NOISE, H-LAG+DCM and H-Lag) in Fig. 11, and move the results for DM and H-LAG+DCM to main text.
>
> | Method| Score   |
> |--------------------|--------:|
> | $\texttt{DM}$ (renamed to FORL-DM)| **27.4** |
> | $\texttt{H-Lag-DCM}$ | 26.3|
> | $\texttt{DQL}$| 10.9|
> | $\texttt{Med-DCM}$ | 17.7|
> | $\texttt{Med-Noise}$| 15.5|
> | $\texttt{H-Lag}$| 17.3|
>
> If past offsets are available, our modular design allows leveraging zero‑shot foundation models with DCM, so improvements in these models will directly benefit FORL. We will report results for all datasets and environments in a dedicated table for methods that do not use past historical ground truth offsets. Furthermore, we have extended  (MOD, Running-Mean,Running-Mean-p and Random Sample Consensus (RANSAC)) in maze2d‑medium environments with all real-data-A,B,C,D,E over five random seeds, which we will include in the supplementary.
>
> - MAD: computes the coordinate-wise median of the differences between the observation and each denoiser prediction, discards any sample whose absolute deviation from this median exceeds $\epsilon$ x Median Absolute Deviation (MAD), takes the median of the remaining inliers to obtain a robust offset estimate, and subtracts that offset from the observation
> - Running-$\mu$:computes a numerically stable, global running mean of offsets from DM across all episodes
> - Running-$\mu$-p: Running-$\mu$ per episode $p$
> - RANSAC: calculates the offsets from DM, sets an adaptive per-dimension threshold as $\epsilon$ x MAD, runs RANSAC by repeatedly sampling one offset candidate and choosing the candidate whose inlier set (differences within that threshold) is largest, averages those inliers to estimate offset.
>
> | Method   | Score   |
> |--------------------|--------:|
> | FORL-DM ($\texttt{DM}$ in main paper)  | **57.1** |
> | FORL-DM-MAD| 45.6  |
> | FORL-DM-Running-$\mu$ |42.9 |
> | FORL-DM-Running-$\mu$-p | 45.5  |
> | FORL-DM-RANSAC   | 47.6  |
>
> > L108 "FORL makes informed decisions at the start of each episode" does not seem like a contribution
>
> Using a zero-shot forecasting foundation model, our agent has an information on the unknown non-stationarity of the new episode, unlike methods that start "blind" and must learn from scratch within the episode. We will instead list FORL-DM in our contributions.
>
> > L240: Maybe use "diffusion" and "timeseries" instead of "multimodal" and "unimodal"
>
> We agree with the reviewer's suggestion, and will use D_(diffusion) and D_(timeseries) for clarity.
>
> > background material, such as the explanation of diffusion models.
> > Equation (5) and (6) appear in the appendix.
>
> To the best of our knowledge, no prior work trains a diffusion model on sequences of $\Delta s$ and actions to generate multimodal candidate states, a design choice that substantially enhances our model’s robustness. To clarify the training procedure, we explicitly detail the conditioning variables and the training procedure. Following reviewer's suggestion we will move the core components of the diffusion model to a dedicated background section, which will also include Equations (5) and (6) previously in the Appendix.

---

> ### Comment · Reviewer_VWZ4 · 2025-08-02
>
> Thank you for your rebuttal.
>
> > The underlying policies (TD3+BC, RORL, Diffusion-QL), and the forecasts predicted by the Zero-Shot FM are identical policy checkpoints for both our method and the baselines
>
> This largely resolves my concern about the lack of hyper-parameter tuning, although tuning RORL should still be beneficial, it should be orthogonal to the proposed method.
>
> > Q: Is it possible to replace this with the output of the diffusion model for prior episodes  A: We investigated this exact scenario with our H‑LAG+DCM, (L1140)
>
> This very nicely addresses my concern, I would highly recommend moving this to the main part of the paper.
>
> >DCM Model support  / MAX-baseline
>
> Thank you for pointing me to the MAX-baseline. It is rather surprising that it performs so poorly, could you provide any additional interpretation/explanation for why/when MAX fails and why/when DCM performs better?
>
> It might be helpful to add some examples of the predictions of the baselines as done in Figure 6, although it is not possible during the rebuttal.

---

> > ### Author Response · Authors · 2025-08-04
> >
> > We sincerely thank the reviewer for their thoughtful feedback and taking the time to engage in discussion. Your suggestions have been invaluable in helping us strengthen our work.
> >
> > >This largely resolves my concern about the lack of hyper-parameter tuning, although tuning RORL should still be beneficial, it should be orthogonal to the proposed method.
> >
> > We are glad this resolves your concern, and we agree that further tuning of RORL is orthogonal to our contributions.
> >
> > >This very nicely addresses my concern, I would highly recommend moving this to the main part of the paper.
> >
> > Thank you for your valuable suggestion. We will move the H-LAG+DCM and FORL-DM to the main text, as it significantly strengthens our contribution regarding the practicality of our method without relying on past ground truth offsets.
> >
> > >Thank you for pointing me to the MAX-baseline. It is rather surprising that it performs so poorly, could you provide any additional interpretation/explanation for why/when MAX fails and why/when DCM performs better?
> >
> > Thank you for raising this important point. **MAX** can fail when the forecast mean of $D_{timeseries}$ is biased, misleading it to select a candidate from a geometrically distant mode that appears more likely under an inaccurate forecast. In contrast **DCM**, succeeds because its state estimation is not dependent on the forecast's mean, but on a non-parametric search for the forecast sample with the highest score (minimum dimension-wise distance). Hence, DCM's prediction error is governed by the accuracy of the forecast sample in $D_{timeseries}$ with the best score. Empirically this yields lower maximum and mean errors compared to **MAX**. Under ideal sampling conditions, DCM samples from the product distribution of these models (Fig. 4). The timeseries forecaster can generate a large set of samples that can be systematically biased, which is why we observe that the DQL(Mean+Lag) and DQL(Median+Lag) also have high maximum prediction error.
> >
> > For a more rigorous analysis of prediction error, we replicated the experimental setup from Figure 6. We then compared the maximum, minimum, and mean prediction error values over the test episodes. The results show significantly stable prediction errors ($\texttt{Maximum Error}\downarrow$:**2.40**) for **FORL(DCM)**, for both maximum error and mean error compared to **FORL(MAX)**  ($\texttt{Maximum Error}\downarrow$:**9.33**) demonstrating its robustness.
> >
> > | Algorithm | $\texttt{Minimum Error}\downarrow$|  $\texttt{Maximum Error}\downarrow$ | $\texttt{Mean Error}\downarrow$ |
> > | :--- | :--- | :--- | :--- |
> > | **FORL(DCM)** | 0.02 | **2.40** | **0.87 ± 0.60** |
> > | **FORL(MAX)** | **0.01** | 9.33 | 2.05 ± 1.74 |
> > | **DQL**| 2.26 | 11.30 | 5.51 ± 2.13 |
> > | **FORL-DM** (*no past offsets*)| **0.01** | 9.94 | 3.68 ± 2.25 |
> > | **DQL(Mean+Lag)** | 0.04 | 6.28 | 1.76 ± 1.13 |
> > | **DQL(median+Lag)** | 0.05 | 6.56 | 1.87 ± 1.19 |
> > | **DMBP+LAG** | 0.03 | 6.28 | 1.69 ± 1.11 |
> >
> > > It might be helpful to add some examples of the predictions of the baselines as done in Figure 6, although it is not possible during the rebuttal.
> >
> > Thank you for your suggestion, we will include these in our paper. To make the analysis above concrete, here are the numerical coordinates and prediction errors for the Fig.6 (**rotated 90 degrees clockwise**). While for this specific scenario, MAX is close to DCM although worse, we observe that across the test episodes, DCM outperforms MAX in terms of robustness for maximum and mean error.
> >
> > ### Left Maze in Figure 6:
> >
> > The true state is at `[2.54, 5.01]`, and the observation is at `[4.97, 6.56]`.
> >
> > | Method | *x,y positions* | $\texttt{Prediction Error}\downarrow$ |
> > | :--- | :--- | :---: |
> > | **FORL (DCM)** | `[2.84, 4.87]` | **0.33** |
> > | FORL(MAX) | `[2.84, 4.85]` | 0.34 |
> > | FORL-DM | `[2.87, 4.84]` | 0.37 |
> > | DQL(Mean+Lag) | `[0.61, 4.17]` | 2.11 |
> > | DQL(Median+Lag) | `[0.40, 4.14]` | 2.32 |
> > | DMBP+LAG | `[0.56, 4.17]` | 2.15 |
> >
> >
> > ### Right Maze in Fig.6
> >
> > The true state is at `[1.33, 4.18]`, and the observation is at `[3.75, 5.73]`.
> >
> > | Method | *x,y positions* | $\texttt{Prediction Error}\downarrow$ |
> > | :--- | :--- | :---: |
> > | **FORL (DCM)** | `[1.62, 3.95]` | **0.37** |
> > | FORL(MAX) | `[1.59, 3.87]` | 0.41 |
> > | FORL-DM | `[1.60, 3.87]` | 0.42 |
> > | DQL(Mean+Lag) | `[0.47, 3.37]` | 1.19 |
> > | DQL(Median+Lag)|`[0.40, 3.31]`| 1.28  |
> > | DMBP+LAG | `[0.40, 3.91]` | 0.97 |
> >
> > We thank the reviewer again for the highly constructive and insightful suggestions.

---

> > ### Author Response · Authors · 2025-08-04
> >
> > We will add the following results to our paper, comparing the normalized scores (mean±std) of our $\texttt{DCM}$ module against the $\texttt{MAX}$ baseline. The table shows that $\texttt{FORL(DCM)}$ has high performance across nearly all settings, with the exception of antmaze-large-diverse, *real-data-E*. The numerical values of prediction error in our Fig.11 are **3.1** for $\texttt{FORL(DCM)}$ and 5.2 for $\texttt{MAX}$.
> >
> > | Environment | *Dataset* | $\texttt{FORL(DCM)}$ | $\texttt{MAX}$ |
> > | :--- | :--- | :---: | :---: |
> > | $\texttt{maze2d-medium}$ | *real-data-A* | **63.3 ± 6.7** | 41.2 ± 8.2 |
> > | $\texttt{maze2d-medium}$ | *real-data-B* | **66.5 ± 18.2** | 58.9 ± 14.1 |
> > | $\texttt{maze2d-medium}$ | *real-data-C* | **86.3 ± 15.7** | 66.1 ± 16.4 |
> > | $\texttt{maze2d-medium}$ | *real-data-D* | **103.4 ± 11.9** | 44.4 ± 21.6 |
> > | $\texttt{maze2d-medium}$ | *real-data-E* | **51.2 ± 13.7** | 11.8 ± 5.5 |
> > | $\texttt{maze2d-medium}$ | **Average** | **74.1** | 44.5 |
> > | --- | --- | --- | --- |
> > | $\texttt{maze2d-large}$ | *real-data-A* | **42.9 ± 4.1** | 11.1 ± 2.3 |
> > | $\texttt{maze2d-large}$ | *real-data-B* | **34.9 ± 9.2** | 28.3 ± 7.1 |
> > | $\texttt{maze2d-large}$ | *real-data-C* | **45.6 ± 4.1** | 34.6 ± 13.6 |
> > | $\texttt{maze2d-large}$ | *real-data-D* | **58.4 ± 6.5** | 18.4 ± 9.9 |
> > | $\texttt{maze2d-large}$ | *real-data-E* | **12.0** | 9.2 ± 6.1 |
> > | $\texttt{maze2d-large}$ | **Average** | **38.8** | 20.3 |
> > | --- | --- | --- | --- |
> > | $\texttt{antmaze-umaze-diverse}$ | *real-data-A* | **65.3 ± 8.7** | 59.7 ± 9.7 |
> > | $\texttt{antmaze-umaze-diverse}$ | *real-data-B* | **74.2 ± 10.8** | 65.0 ± 10.5 |
> > | $\texttt{antmaze-umaze-diverse}$ | *real-data-C* | **78.8 ± 8.5** | 76.2 ± 5.4 |
> > | $\texttt{antmaze-umaze-diverse}$ | *real-data-D* | **75.8 ± 8.0** | 63.3 ± 6.8 |
> > | $\texttt{antmaze-umaze-diverse}$ | *real-data-E* | **81.3 ± 6.9** | 72.0 ± 15.0 |
> > | $\texttt{antmaze-umaze-diverse}$ | **Average** | **75.1** | 67.2 |
> > | --- | --- | --- | --- |
> > | $\texttt{antmaze-medium-diverse}$ | *real-data-A* | **44.0 ± 7.9** | 36.0 ± 4.8 |
> > | $\texttt{antmaze-medium-diverse}$ | *real-data-B* | **55.8 ± 7.0** | 36.7 ± 7.5 |
> > | $\texttt{antmaze-medium-diverse}$ | *real-data-C* | **52.9 ± 9.5** | 37.1 ± 3.7 |
> > | $\texttt{antmaze-medium-diverse}$ | *real-data-D* | **64.2 ± 8.6** | 37.5 ± 6.6 |
> > | $\texttt{antmaze-medium-diverse}$ | *real-data-E* | **26.7 ± 4.7** | **26.7 ± 7.1** |
> > | $\texttt{antmaze-medium-diverse}$ | **Average** | **48.7** | 34.8 |
> > | --- | --- | --- | --- |
> > | $\texttt{antmaze-large-diverse}$ | *real-data-A* | **34.3 ± 5.7** | 21.7 ± 7.9 |
> > | $\texttt{antmaze-large-diverse}$ | *real-data-B* | **46.7 ± 11.9** | 20.0 ± 6.8 |
> > | $\texttt{antmaze-large-diverse}$ | *real-data-C* | **33.8 ± 6.8** | 23.8 ± 6.4 |
> > | $\texttt{antmaze-large-diverse}$ | *real-data-D* | **46.7 ± 12.6** | 21.7 ± 7.5 |
> > | $\texttt{antmaze-large-diverse}$ | *real-data-E* | 11.3 ± 7.3 | **20.7 ± 4.9** |
> > | $\texttt{antmaze-large-diverse}$ | **Average** | **34.6** | 21.6|

---

> > > ### Comment · Reviewer_VWZ4 · 2025-08-05
> > >
> > > Thank you for providing the additional results concerning the MAX baseline.
> > >
> > > My concerns about DCM have mostly been addressed, but I would recommend to revise the presentation in the paper to convey why it was chosen over the (perhaps more natural) Bayes KDE and MAX methods.

---

> ### Author Response · Authors · 2025-08-05
>
> Thank you for your highly constructive feedback and for taking the time to engage with our work. We will clarify the limitations of standard approaches like MAX and Bayes KDE compared to **DCM** in the **main paper**, adding the analysis of maximum and minimum prediction errors. Additionally, we will include detailed numerical tables for the average scores in Appendix Fig. 11, and Bayes KDE. Please let us know if you have any further questions, and thank you again for your excellent suggestions.

---

> ### Author Response · Authors · 2025-08-09
> **Manipulation task in OGBench**
>
> To address your concerns about settings "outside maze-like" environments, we share detailed results from the offline RL **manipulation task** ($\texttt{cube-single-play-singletask}$) in the recent **OGBench** [1] benchmark (ICLR2025), using five random seeds. We use the recent **Flow Q-Learning (FQL)** [2] (ICML2025) as the policy for all methods with identical policy checkpoints with the official hyperparameters from its open-source repository; FQL has achieved strong performance on this task. We inject non-stationarity into the dimensions corresponding to all joint positions/velocities and end-effector position/yaw from the same domains $\texttt{real-data-}{A,B,C,D,E}$ (GluonTS).$\mathbf{FORL(DCM)-F}$ consistently outperforms the baselines on the manipulation task, indicating robustness. Consistent with our previous results we discuss our findings below.
>
> - **FORL State Estimation**: DCM component outperforms MAX and BayesKDE. $\mathbf{FORL(DCM)-F}$ corresponds to **our proposed framework** $\mathbf{FORL-F}$ that includes DCM.
> - **FORL Diffusion Model**: Although $\mathbf{FORL-DM-F}$ **does not have access to past ground truth offsets**, it performs similar to   $\mathbf{FQL+Lag-\bar{s}}$ that **has** access to past ground truth offsets. These results indicate the importance of our proposed diffusion model component dedicated to state generation for zero-shot deployment.
>
> | Dataset | Time Series Dataset | $\mathbf{FQL}$ |  $\mathbf{FORL-DM-F}$ |  $\mathbf{FQL+Lag-\bar{s}}$ |  $\mathbf{FORL(MAX)-F}$ |  $\mathbf{FORL(Bayes KDE)-F}$ |  $\mathbf{FORL(DCM)-F}$ |
> | :--- | :--- | :---: | :---: | :---: | :---: | :---: | :---: |
> |  $\texttt{cube-single-play}$  | $\texttt{real-data-A}$ | $0.0 ± 0.0$ | $3.0 ± 1.4$ | $0.0 ± 0.0$ | $0.0 ± 0.0$ | $0.0 ± 0.0$ | $\textbf{23.7 ± 3.6}$ |
> | $\texttt{cube-single-play}$  | $\texttt{real-data-B}$  | $0.0 ± 0.0$ | $6.7 ± 5.6$ | $15.0 ± 7.0$ | $43.3 ± 4.8$ | $2.5 ± 2.3$ | $\textbf{60.0 ± 7.0}$ |
> | $\texttt{cube-single-play}$  | $\texttt{real-data-C}$  | $0.4 ± 0.9$ | $4.6 ± 1.7$ | $10.0 ± 1.7$ | $39.6 ± 4.4$ | $38.3 ± 3.8$ | $\textbf{42.1 ± 5.6}$ |
> |  $\texttt{cube-single-play}$  | $\texttt{real-data-D}$ | $0.0 ± 0.0$ | $2.5 ± 2.3$ | $0.8 ± 1.9$ | $7.5 ± 1.9$ | $0.0 ± 0.0$ | $\textbf{70.0 ± 13.0}$ |
> | $\texttt{cube-single-play}$ | $\texttt{real-data-E}$| $0.0 ± 0.0$ | $8.0 ± 3.0$ | $0.0 ± 0.0$ | $21.3 ± 8.0$ | $0.0 ± 0.0$ | $\textbf{32.7 ± 9.5}$ |
> |$\texttt{cube-single-play}$ | **Average** | $0.1$ | $5.0$ | $5.2$ | $22.3$ | $8.2$ | $45.7$ |
>
>
> [1]Park, Seohong, Kevin Frans, Benjamin Eysenbach, and Sergey Levine. "Ogbench: Benchmarking offline goal-conditioned rl." ICLR 2025
>
> [2]Park, Seohong, Qiyang Li, and Sergey Levine. "Flow q-learning." ICML 2025
>
>
> We will include these results in our paper.
> We thank you again for your inspiring suggestions and taking the time to participate in the discussion.

---

### Note · Authors · 2025-08-14

We are deeply grateful to our reviewers for their insightful feedback, comprehensive reviews and to the Area Chair for overseeing this thorough and constructive discussion. Our problem is recognized as "realistic and underexplored" (cynB), an "important deployment issue" (cynB), "well motivated" (VWZ4), "challenging" (VWZ4, x1rM), and "novel" in its formulation (VWZ4). Our method is "novel and modular" (cynB), "original" (x1rM), and a "technically innovative approach" (x1rM). FORL demonstrates a "clear advantage" (VWZ4), "significantly outperforms relevant baselines" (x1rM). VqtP noted our paper "has a lot of potential and has some interesting, novel results". In response to the reviewers' constructive feedback, we have provided new experiments, clarifications, and analyses to address the points raised.

- Reviewers VWZ4, cynB, x1rM underscored the importance of performance in the no past offset access setting. We have therefore elevated our original results (showing strong performance with FORL-DM & H-LAG+DCM) to the main paper for greater visibility and added four new no-access ablations (VWZ4).

- We natively handle intra-episode non-stationarity (cynB,x1rM) with offsets revealed after each episode FORL 61.5 vs DQL(Mean+Lag) 46.9 and no access FORL-DM 33.0 vs DQL 17.4 (5 seeds).

- We extended DCM and MAX ablations with 3 more baselines (5 seeds) (VWZ4); DCM is superior/competitive and requires no tuning. We added a focused max/mean-error analysis to validate DCM's advantage, and baseline prediction examples.

- We expanded our evaluation with new OGBench (ICLR2025) (VqtP) and more offline RL methods (cynB) IQL and new FQL (ICML2025). To ensure a fair comparison, all methods in every existing and new experiment use identical policy checkpoints. This addressed the hyperparameter tuning concern, which VWZ4 confirmed was "largely resolved".

- Sensitivity to forecasting errors (x1rM). A non-stationarity-wise reanalysis of our existing results shows up to 56.5% error reduction, largest at moderate difficulty; now included in the paper.

- We incorporated all clarity/presentation edits based on the reviewers' constructive feedback.

Reviewer cynB wrote "I will increase my rating to 4" after stating all their concerns were addressed; VWZ4 said concerns were "mostly/largely/very nicely" addressed, requested presentation revisions (implemented); VqtP noted potential, requested clarifications and OGBench (both included); x1rM appreciated our "detailed response".

---

### Decision · Program_Chairs · 2025-09-17

**Decision:**

Accept (spotlight)

**Comment:**

This paper studies nonstationary offline RL in a setting where training data are stationary but deployment involves an unknown, additive offset to the state that varies across episodes while remaining constant within each episode. The authors propose a method combining a diffusion-based learned environment model with a zero-shot time-series model that forecasts future offsets from past ones. Experiments demonstrate clear advantages over strong baselines in terms of both prediction error and achieved reward.

Reviewers agree that this is an important and timely problem and that the proposed techniques are novel and empirically strong. The rebuttal strengthened the case for acceptance by providing additional baselines and experiments in OGBench. There was also constructive discussion on how restrictive some of the assumptions are. Reviewers were broadly convinced by the authors’ arguments.

A main concern shared across reviewers relates to the clarity and structure of the paper. The presentation could be streamlined, and assumptions and limitations—such as the treatment of offsets—should be explained more clearly upfront. I recommend the authors address this in the camera-ready version. Overall, I recommend acceptance.